**Data Availability Statement:** All relevant data are within the manuscript and its Supporting information files.

# Renin angiotensin system genes are biomarkers for personalized treatment of acute myeloid leukemia with Doxorubicin as well as etoposide

**Seyhan Turk** [1]*, **Can Turk**[2], **Muhammad Waqas Akbar**[3], **Baris Kucukkaraduman**[3], **Murat Isbilen**[3], **Secil Demirkol Canli** [4], **Umit Yavuz Malkan** [5], **Mufide Okay**[6], **Gulberk Ucar**[1], **Nilgun Sayinalp**[6], **Ibrahim Celalettin Haznedaroglu**[6], **Ali Osmay Gure**[3]

**1** Department of Biochemistry, Hacettepe University, Ankara, Turkey, **2** Department of Medical Microbiology, Lokman Hekim University, Ankara, Turkey, **3** Department of Molecular Biology and Genetics, Bilkent University, Ankara, Turkey, **4** Molecular Pathology Application and Research Center, Hacettepe University, Ankara, Turkey, **5** Department of Hematology, University of Health Sciences, Ankara, Turkey, **6** Department of Hematology, Hacettepe University, Ankara, Turkey

* seyhan.turk@hacettepe.edu.tr

## Abstract

Despite the availability of various treatment protocols, response to therapy in patients with Acute Myeloid Leukemia (AML) remains largely unpredictable. Transcriptomic profiling studies have thus far revealed the presence of molecular subtypes of AML that are not accounted for by standard clinical parameters or by routinely used biomarkers. Such molecular subtypes of AML are predicted to vary in response to chemotherapy or targeted therapy. The Renin-Angiotensin System (RAS) is an important group of proteins that play a critical role in regulating blood pressure, vascular resistance and fluid/electrolyte balance. RAS pathway genes are also known to be present locally in tissues such as the bone marrow, where they play an important role in leukemic hematopoiesis. In this study, we asked if the RAS genes could be utilized to predict drug responses in patients with AML. We show that the combined *in silico* analysis of up to five RAS genes can reliably predict sensitivity to Doxorubicin as well as Etoposide in AML. The same genes could also predict sensitivity to Doxorubicin when tested *in vitro*. Additionally, gene set enrichment analysis revealed enrichment of TNF-alpha and type-I IFN response genes among sensitive, and TGF-beta and fibronectin related genes in resistant cancer cells. However, this does not seem to reflect an epithelial to mesenchymal transition per se. We also identified that RAS genes can stratify patients with AML into subtypes with distinct prognosis. Together, our results demonstrate that genes present in RAS are biomarkers for drug sensitivity and the prognostication of AML.

**Funding:** This study was supported by The Scientific and Technological Research Council of Turkey (116S350).

**Competing interests:** The authors have declared that no competing interests exist.

## Introduction

Leukemia, lymphoma and multiple myeloma are the three main types of highly heterogeneous hematological malignancies that are derived from myeloid and lymphoid cell lineages [1]. Acute myeloid leukemia (AML) is characterized by abnormal expansion of immature myeloid cells and their accumulation in the bone marrow and blood, interfering with normal cellular growth [2]. AML is a highly aggressive cancer with poor prognosis. It is also the most common type of acute leukemia in adults. Treatment strategies and success rates vary depending on many factors, including the subtype of AML, prognostic factors, age and general health status of the patient [3]. Standard treatment regimens based on patient stratification include the combination of chemotherapeutics such as Cytarabine, Daunorubicin and Etoposide with or without radiotherapy. However, high heterogeneity of clinical outcomes in AML patients suggests that current classifications fail to distinguish patient subgroups sufficiently [4].

A not so well studied protein network in the context of AML is the Renin-Angiotensin System (RAS). RAS is composed of several gene products which play a critical role in regulating blood pressure, renal vascular resistance and the fluid/electrolyte balance [5, 6]. The idea of a local RAS operating independent of the circulating RAS was brought into light by demonstrating localized RAS elements in organs other than liver (angiotensinogen), kidney (renin) and lung (ACE). Localized RAS elements were found in many organs such as the brain, blood vessels and heart [7, 8]. It is predicted that locally produced angiotensins have important homeostatic functions and may contribute to local tissue dysfunction and diseases [8]. The presence of local RAS specific to the hematopoietic bone marrow microenvironment was reported for the first time in 1996 [9]. Major RAS molecules have been identified in the bone marrow microenvironment, such as renin, angiotensinogen, angiotensin receptors and angiotensin converting enzymes (ACEs) [10]. Locally active bone marrow RAS affects important stages of physiological and pathological blood cell production through autocrine, paracrine and intracrine pathways [11, 12]. Local bone marrow RAS peptides control the development of hematopoietic niche, myelopoiesis, erythropoiesis, thrombopoiesis and other cellular lineages [13–19]. Local RAS is also active in the primitive embryonic hematopoiesis phase [20–23]. The presence of renin, ACE, angiotensin II (Ang-II) and angiotensinogen in leukemic blast cells has been demonstrated, and local bone marrow RAS has been shown to play a role in the development of neoplastic malignant blood cells [24–26].

Establishing a role for genes involved in the development and biology of cancers, as prognostic and chemotherapeutic markers, is one of the most effective and successful approach used in the classification of malignancies. Thus, here we aimed to define AML subgroups based on expression of RAS genes. We also aimed to test if the resulting tumor subtypes differ in their responses to drugs and to demonstrate distinct prognostic profiles.

## Materials and methods

### *In silico*

**Datasets.** Cancer Genome Project (CGP) gene expression data (E-MTAB-783) [27, 28] was downloaded from ArrayExpress website (https://www.ebi.ac.uk/arrayexpress/), and drug screening data [29] was downloaded from the CGP database. Microarray dataset GSE12417 [30], corresponding to AML patients, was downloaded from the National Center for Biotechnology Information (NCBI) Gene Expression Omnibus (GEO) database. The training cohort of 163 patients in GSE12417 [30] was used for our analyses since this was the same microarray platform as the one used for CGP.

**Data normalization and variance analysis.**   CGP gene expression data was normalized by the RMA method using the BRB-Array Tools software [31].

In order to choose the RAS genes that would be used in real-time PCR for validation studies, we first aimed to choose the most variable genes that would likely give detectable fold differences *in vitro* by PCR. Analysis of variance was performed using all 39 probesets corresponding to the 25 genes in the RAS and genes with at least 0.8 of variance. Above these thresholds, the mean expression was at least 5.5 and the log fold difference between min to max was above 3 for all probesets. Thus, nine probesets corresponding to eight genes (*CTSG, CPA3, AGT, ANPEP, IGF2R (two probesets), RNPEP, ATP6AP2 and CTSA*) were selected to be used in further analyses (S1 Table).

**IC50 calculation methods.**   In order to calculate drug response parameters such as IC50, EC50, activity area and Amax, the growth rate of the cells were depicted as a function of drug concentration by being modeled with non-linear logistic regression as explained in De Lean *et. al* [32], which is also reported in NIH/NCGC assay guidelines [33]. While the non-linear logistic regression function used to model data is used widely for cytotoxicity calculations, here for the first time we used six different versions of this function and selected the one with the lowest standard error rate among all for the calculation of cytotoxicity values. We name this approach the 6-model (6M).

Thus, six different models were derived from the following non-linear logistic regression function:

$$Y = (a - d/(1 + (X/c)^b) + d)$$

where *Y* is the percent growth of the cells, *X* is the arithmetic drug concentration, *a* is the percent growth of the cells when the cells are not treated with the drug, *d* is the percent growth of the cells for infinite dose, i.e. a dose for which there is no additional effect when increased, c is the dose corresponding to percent growth exactly between *a* and *d*, and *b* is the Hill slope factor that is used to define the steepness of the curve fitted.

The following are the conditions required for the generation of 6-models:

1. **3-Parameter model:** Curves were fitted without using Hill slope factor *b*.

2. **3-Parameter Top 100 model:** Curves were fitted without using Hill slope factor *b* and with *a* = 100.

3. **3-Parameter Bottom 0 model:** Curves were fitted without using Hill slope factor *b* and with *d* = 0.

4. **4-Parameter model:** Formula is used as it is.

5. **4-Parameter Top 100 model:** Curves were fitted with *a* = 100.

6. **4-Parameter Bottom 0 model:** Curves were fitted with *d* = 0.

Six different drug response parameters are calculated out of the fitted curves as follows:

- **IC50:** Value of *X* when $\hat{Y}$ = 50%

- **IC90:** Value of *X* when $\hat{Y}$ = 90%

- **IC95:** Value of *X* when $\hat{Y}$ = 95%

- **EC50:** Value of *X* when $\hat{Y}$ = *a*+*d*

- **Amax:** *a* − *d*

- **Activity Area:** $\Sigma \hat{Y}X$, (sum of $\hat{Y}$s for each 0.01 increment of $X$ fitted), where $\hat{Y}$ is the predicted value of $Y$ by the curve fitted.

With the 6M approach we recalculated IC50 values that were also included in the raw CGP data for the 17 AML cell lines treated with four drugs (ATRA, Cytarabine, Etoposide and Doxorubicin) using an in-house R script "*SixModelIC50 V3.r*" (https://github.com/muratisbilen/6-Model_IC50_CalculationV3.git).

These drugs were selected as we obtained AML chemotherapy treatment protocols from the Department of Hematology, Hacettepe University and compiled a list for all drugs in these protocols. Among these only ATRA, Cytarabine, Etoposide and Doxorubicin were present in the CGP database.

We referred to the recalculated IC50 data as 6M IC50 and performed a Pearson r correlation analysis between CGP IC50s and recalculated 6M IC50s to test the compatibility.

In addition, IC50 values were calculated using the 6M approach on the data obtained from in vitro analysis in which nine AML cell lines were treated with Doxorubicin and Etoposide.

**Linear regression analyses.** We performed correlation analysis between expression values of the eight genes and drug data (both CGP IC50 data and 6M IC50 data) individually. To identify if multiple genes can be used to better identify the relationship between gene expression and drug sensitivity data, linear regression analyses were performed using the Minitab 17 software (https://www.minitab.com). Seventeen AML cell lines from the CGP database were either randomly divided into two groups, the discovery group (12 cell lines) and the test group (five cell lines), or chosen manually so that the sensitivity range of cells in both groups spanned as large variance as possible. To generate a linear regression model for each drug (ATRA, Cytarabine, Etoposide, Doxorubicin), IC50s of the discovery cell line group obtained either from CGP or recalculated as 6M IC50, and expression of the eight RAS genes which were selected from variance analysis, were used as predictors. As a measure of the response variable variation explained by each linear regression model, we used the adjusted (adj.) $R^2$ values. To test consistency of the linear regression models generated with the eight genes, we replicated the random divison of groups ten times and reported the average of the adjusted $R^2$.

Furthermore, to identify a minimal gene list for the prediction of chemosensitivity, the discovery group was used to fit a model explaining the drug response using "best subsets" function of the software, which runs all possible regression models with one variable, two variables and so on, based on a list of predictors, enabling the user to choose a smaller set of predictors that can explain the response. The subset with the highest $R^2$ (adj.) was selected as the best model. Regression formula of the best models ($y = \pm a + [n1 \times x1] \pm [n2 \times x2] \pm [n3 \times x3] \pm [n4 \times x4] \ldots$) were applied for the test group of each drug. In the regression formula $y$ (predicted IC50 values) were calculated where $a$ and $n$ are the constant values, $x$: gene expression values of the 12 cell lines in the discovery group. Also, the goodness of fit measure Sy.x were computed by Graphpad. Sy.x is a standard deviation of the residuals that here has been used to describe the difference in standard deviations of CGP IC50 and 6M IC50 versus predicted IC50s. It is a goodness-of-fit measure used to show how well our predicted IC50s fit with CGP and 6M IC50 values. All the correlations were calculated with Graphpad software as Pearson's r and p values.

**Hierarchical clustering analysis.** Cluster 3.0 (http://bonsai.hgc.jp/~mdehoon/software/cluster/software.htm) [34] and Java Treeview (http://jtreeview.sourceforge.net/) [35] software were used for hierarchical clustering analysis with mean standardized gene expression values for each dataset. Hierarchical clustering was performed by clustering both genes and arrays using Euclidian distance as similarity metric and complete linkage as clustering method.

**Gene sets enrichment analysis—GSEA.** Gene set enrichment analysis was performed using the GSEA guideline (https://www.gsea-msigdb.org/gsea/index.jsp) [36].

Briefly, dataset E-MTAB-783 [27, 28] has 22277 probesets IDs and these were collapsed into 13321 genes. For genes with more than one probeset, one with the highest expression was selected. "C5_all Gene ontology v6.1 database" was used for the analysis which has gene sets that contain genes annotated by the same GO term. We used default filtering criteria in GSEA for gene set sizes, which includes genesets with sizes between 15–500. After applying this filter, analysis was performed for 4081 gene sets.

**Mutation analyses.** Mutation data of AML cell lines was downloaded from Genomics of Drug sensitivity in Cancer database (https://www.cancerrxgene.org/downloads/bulk_download) [37]. 14 out of the 17 AML cell lines used in our analyses were available. Seven genes which were mutated in at least three AML cell lines were analyzed further.

## In vitro

**Cell lines and cytotoxicity experiments.** HEL92.1.7 (2111706), and QIMR-WIL (86030601) cell lines were purchased from Sigma Aldrich (St. Louis, Mo., USA), KASUMI-3 (CRL-2725), GDM-1 (CRL-2627) and CESS (TIB-190) cell lines were purchased from ATCC (Virginia, USA) and P31/FUJ (JCRB0091), NOMO-1 (IFO50474), KASUMI-1 (JCRB1003) and SKM-1 (JCRB0118) cell lines were purchased from JCRB Cell Bank (Osaka, Japan). Cell lines were authenticated by manufacturers, all cell lines were morphologically checked by microscope and routine mycoplasma testing was performed by PCR. HEL92.1.7, GDM-1, CESS, P31/FUJ and NOMO-1 were cultured and maintained in RPMI-1640 medium (Sigma-Aldrich, R0883 (St. Louis, Mo., USA)) supplemented with 10% fetal bovine serum (FBS) (Sigma-Aldrich, F6178 (St. Louis, Mo., USA)), 1% penicillin-streptomycin (Sigma-Aldrich, 11074440001 (St. Louis, Mo., USA)), and 1% 200 mM L-glutamine (Sigma-Aldrich, G7513 (St. Louis, Mo., USA)). KASUMI-1, SKM-1 and KASUMI-3were cultured in RPMI-1640 medium but with 20% FBS. QIMR-WIL was cultured in DMEM medium (Sigma-Aldrich, D6546 (St. Louis, Mo., USA)) but with 10% FBS, 1% penicillin-streptomycin, and 1% 200 mM L-glutamine. All cell lines were cultured at 5% $CO^2$ and 37 ˚C in a humidified incubator.

Doxorubicin (D1515) and Etoposide (E1383) were purchased from Sigma-Aldrich (St. Louis, Mo., USA) and were dissolved in DMSO. Cell viability was measured using CellTiter-Glo reagent (G7572, Promega, Fitchburg, Wisconsin, USA). 7000 cells/well in 90 μl medium were plated in each well of a 96-well plate. Cells were treated with six different concentrations of Doxorubicin or Etoposide separately (20, 10, 2, 1, 0.2, 0.1 μM). After 72 hours of drug treatment, cells were treated with CellTiter-Glo reagent and the luminescence signal was then recorded with a microplate luminometer (Turner Designs, CA, USA). All drug treatment experiments were repeated three times. Growth percentages were calculated for each drug and cell line, and cytotoxicity values were calculated using the 6M approach.

**qRT-PCR.** *AGT*, *ANPEP*, *ATP6AP2*, *CPA3*, *CTSA and IGF2R* genes' expression was quantified using SYBR ™ Green master mix (Bio-Rad, #1725150, (USA)). PCR reactions were run under cycling conditions according to manufacturer's instructions. GAPDH was used as a reference gene in all reactions. qRT-PCR relative gene expression data was calculated using ddCT method [38].

Using qRT-PCR relative gene expression data, predicted IC50 values were calculated with the formulas generated by linear regression analyses of *in silico* data using qRT-PCR based expression values as predictors. Primers used in this study are shown in Table 1. GAPDH was used as endogenous control.

**Table 1. Primer sequences are for selected genes.**

| Gene Name | Forward Primer | Reverse Primer |
|---|---|---|
| AGT | GGCCAGCAGCAGATAACAACC | AACTGGGAGGTGCATTTGTGC |
| ANPEP | CGTTCTCTCTGCCTGTGAGC | AGGCCGTTCATTGTCCATCG |
| ATP6AP2 | GATCCTTGTTGACGCTCTGC | CTTGCTGGGTTCTTCGCTTG |
| CPA3 | TGCCCTCTGTTTGGAATAAGCC | GCTGGGTCCAAACTTCACTTGG |
| CTSA | CTCTACCGAAGCATGAACTCCC | TACTTCACTAACCAGGGCCG |
| IGF2R_probe1 | CTCCCACCCAGTGAGAAACG | TCGTCATGGAAGGACACCAG |
| IGF2R_probe2 | GGTGTTCTTATTCTGGCGGC | CAAACAAGCCAGCCAAACCG |
| GAPDH | GGAGCGAGATCCCTCCAAAAT | GGCTGTTGTCATACTTCTCATGG |

## Clinical data validation

**Log Rank with Multiple Cutoffs (LRMC) and survival analysis.** In regression analysis, four formulas were generated for Doxorubicin and Etoposide using both CGP and 6M IC50 data. *IGF2R*, *CTSA*, *ATP6AP2* are common in three of the four formulas except for 6M IC50 data for Etoposide. Therefore, these genes were chosen to test relationships with clinical outcome.

Clinical data were obtained from the training cohort of the GSE12417 [30] dataset (AML Cooperating Group 1999). In the AMLCG 1999 cohort, patients were treated with TAD: Thioguanine, Cytarabine and Daunorubicin, or HAM: Cytarabine, Mitoxantrone protocols followed by the TAD protocol. We used an in-house R script (https://github.com/muratisbilen/LRMC.git) (Log Rank Multiple Cutoff, LRMC) by which log-rank test-based p-values associated with hazard ratio (HR) could be obtained using all possible cutoff values representing each sample in a given dataset and best cutoff is selected as in [39, 40]. Using this approach, we selected best cutoffs for *IGF2R*, *ATP6AP2 and CTSA* genes to be used for clinical correlation studies and Kaplan-Meier plots.

Patients with gene expression lower than cutoff, for each gene individually, were labeled as 'Low' (low expression) and higher than cutoff were labelled as 'High' (high expression). Univariate cox regression analyses were performed and Kaplan-Meier graphs were drawn using SPSS Statistics 19 (IBM, Chicago, IL, USA).

Additionally, the expression of all these three genes (*IGF2R*, *CTSA* and *ATP6AP2)* was evaluated together as good and bad prognostic groups. Patients were grouped as "Good" if they have high expression levels of IGF2R and CTSA and low expression levels of ATP6AP2 defined by expression value cutoffs in previous analysis. Rest of the patients were grouped as "Bad". Then Kaplan Meier survival analysis was performed for these groups.

## Results

### Discovery of RAS drug sensitivity biomarker genes

The RAS consists of the 25 genes, corresponding to 39 probesets in Affymetrix HG-U133A, a microarray platforms used in the Cancer Genome Project (CGP) [27, 28]. For the 17 AML cell lines, both drug cytotoxicity and gene expression data are available in the CGP database [27, 28]. We focused only on genes which showed high variation in expression for further validation and therefore, selected nine probesets (eight genes) as described in the methods section (S1 Table). We recalculated IC50 values using the 6M approach applied to raw CGP cytotoxicity data (see Materials and methods). Using Pearson correlation we observed strong correlations between CGP IC50 and 6M IC50 for drugs widely used in AML (Cytarabine, Etoposide, Doxorubicin) but not for ATRA (S2 Table). To identify biomarkers of chemosensitivity, we

calculated Pearson correlation between gene expression and IC50 values obtained from CGP and generated by 6M approach. We thus identified six gene/drug cytotoxicity correlations which were significant with CGP IC50 values, and seven significant correlations with 6M IC50 values. Four gene/drug associations were common to both analyses (S3 Table). Linear regression analysis was then performed to test whether the combined expression analyses of genes could correlate better with drug sensitivity data or not. Thus, we generated discovery and test groups. Each group include a wide range of cell line IC50 values as possible. Linear regression models for drug sensitivity prediction were generated for the discovery group (12 cell lines) using expression data of highly variant eight genes and IC50 values obtained from CGP and 6M IC50 of four drugs in Minitab 17. Then, obtained results tested with the validation group (five cell lines). The models generated with combined expression analyses of the eight genes resulted in high $R^2$ (adj) values for Etoposide, Doxorubicin and Cytarabine but no model could be generated for ATRA (S4 Table). As independent datasets with drug sensitivity data for these compounds do not exist, we utilized a cross-validation method to test the robustness of the proposed models by generating the discovery and test groups 10 times, with 12 and five cell lines, respectively. The average of 10 $R^2$ values generated from discovery groups was calculated for both CGP and 6M IC50s. Our results showed that the 10 random models of sensitivity to Doxorubicin had an average $R^2$ above 85% for both CGP and 6M IC50s, but $R^2$ decreased slightly for models of sensitivity to Etoposide while $R^2$ values highly decreased for models of sensitivity to Cytarabine (S4 Table) when compared to those generated for cell lines that were manually selected. We therefore, focused on Doxorubicin and Etoposide for further analyses.

We then aimed to identify the minimal number of genes that needed to be included in combinations into the models that would give the highest correlation using the 'best subsets function' of Minitab. The software selected three genes/probesets for Doxorubicin when either CGP and 6M IC50 values were used and, four and five genes/probeset combinations for Etoposide using CGP and 6M IC50 values, respectively; all together corresponding to a total of six genes (*AGT*, *ANPEP*, *ATP6AP2*, *CPA3*, *CTSA and IGF2R (two probesets)*) (S5 Table), when the analysis was performed with the discovery group. Applying the resulting models to the test group showed the reliability of all models. As shown in Fig 1, the goodness of fit measures (R sq. and Sy.x) were 0.9 and 0.21 for Doxorubicin as modeled using 6M IC50 data and 0.89 and 0.34 when we used CGP IC50 values. Similarly, for Etoposide, these two measures were 0.78 and 0.34 for 6M IC50 and 0.77 and 0.57 for CGP IC50 values.

### *In vitro* validation of biomarker genes

We next asked if the linear regression models generated *in silico* could predict *in vitro* cytotoxicity. For this purpose, we determined gene expression values by qRT-PCR for the six RAS genes (*AGT*, *ANPEP*, *ATP6AP2*, *CPA3*, *CTSA and IGF2R (two probesets)*) and used these to predict *in vitro* IC50 values obtained for Etoposide and Doxorubicin calculated with 6M approach for nine AML cell lines (see Materials and methods section). Correlation analysis showed that *in silico* and *in vitro* gene expression data were highly concordant except for CTSA (r: >0.7 and p-value <0.05) (S6 Table). We applied normalized gene expression values obtained in vitro to the *in silico* generated linear regression models (using four regression formulas) (S7 Table). Thus, utilized linear regression formulas with qRT-PCR gene expression data showed a good correlation with *in silico* data for Doxorubicin but not Etoposide (Table 2).

### Biological features of drug sensitive and resistant cells

Cell lines sensitive to Etoposide and Doxorubicin were almost identical (S1 Fig). To determine molecular mechanisms underlying differential response to Etoposide and Doxorubicin, we

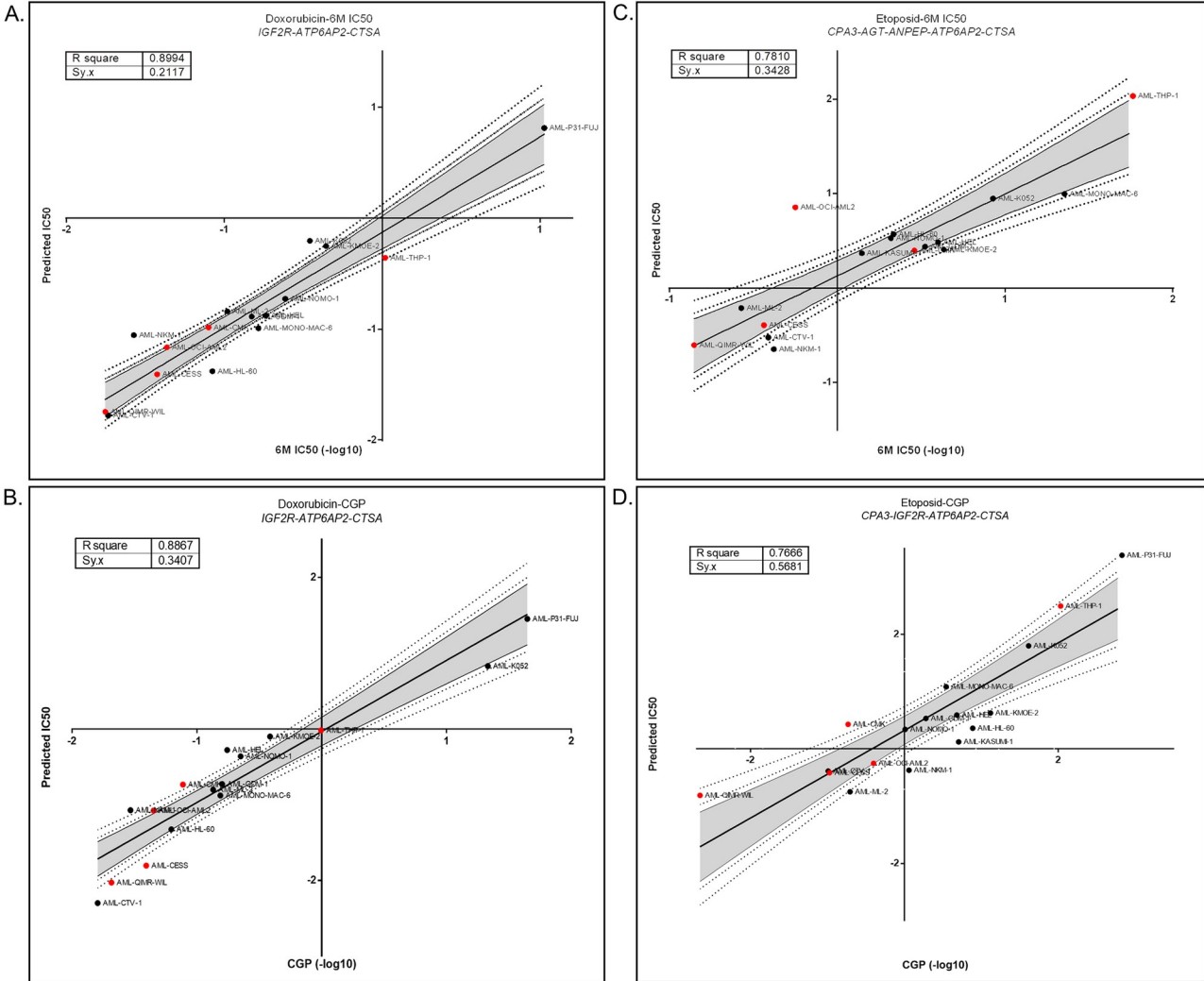

**Fig 1. Reliability of Doxorubicin and Etoposide sensitivity predictions in linear regression models generated using the 12 AML cells.** Linear regression models were generated using the discovery group and applied to the test group to predict sensitivity values. Reliability of sensitivity predictions was measured with goodness of fit test for Doxorubicin 6M IC50 (A) resulting 0.9 R sq. and 0.21 Sy.x Doxorubicin CGP IC50 resulting (B) 0.89 R sq. and 0.34 Sy.x Etoposide 6M IC50 resulting (C) 0.78 R sq. and 0.34 Sy.x Etoposide CGP IC50 resulting (D) 0.77 R sq. and 0.57 Sy.x with 90, 95, and 99% confidence intervals. Black dots represent cell lines used for discovery group, and red dots for the test/validation group.

**Table 2. Pearson's correlation analysis between *in vitro* 6M IC50 values and predicted IC50 values from CGP / 6M IC50 linear regression formulas.**

| *Applied formulas* | **Pearson's r** | ***p*-value** |
|---|---|---|
| Etoposide (CGP) | 0.1271 | 0.7446 |
| Etoposide (6M IC50) | -0.0579 | 0.8825 |
| Doxorubicin (CGP) | 0.7107 | **0.0319** |
| Doxorubicin (6M IC50) | 0.6925 | **0.0387** |

Predicted IC50 values obtained from linear regression formulas generated with 6M IC50 and CGP IC50 values showed high correlation with in vitro IC50s obtained from cytotoxicity experiments for Doxorubicin but not for Etoposide. For prediction of IC50s, normalized qRT-PCR gene expression values were used in the linear regression formulas.

performed gene set enrichment analyses (GSEA) with sensitive and resistant subgroups for Gene Ontology (GO) gene sets. Several gene sets were significantly enriched among sensitive and resistant cell lines (FDR q-value<0.25). Gene sets enriched in sensitive cells with a FDR q-value of lower than 0.25 included TNF-receptor interacting, and response to type I IFN stimulus; while gene sets such as regulation of TGF-beta production and FN-binding were enriched in resistant cells suggesting a mesenchymal phenotype (S2 Fig and S8 Table).

To determine if the differentially expressed genes could be reflecting Epithelial-Mesenchymal Transition (EMT), we compared E-Cadherin (epithelial marker) and Vimentin (mesenchymal marker) expression using t-test, between sensitive and resistant cell groups. EMT is the process that epithelial cells lose the apical-basal polarity and cell adhesion, and transform to invasive mesenchymal cells [41]. It is known to play an important role in biological and pathological processes such as cancer progression, metastasis and drug resistance [42–44]. In our analysis, E-cadherin and Vimentin expression were not significantly different between sensitive and resistant groups defined in S1 Fig (p>0.1) (S3 Fig).

Then, in order to understand whether the mutational profile is involved in sensitivity to Doxorubicin, we analyzed mutational data of sensitive, intermediate and resistant groups of AML cell lines (see Materials and methods). Although we have a small sample size, especially in the resistant group (n = 2), we observed that both of the resistant cell lines are NRAS and P53 mutant, whereas all of the sensitive cell lines (n = 5) were wild type for these genes. However, these results need to be validated in larger sample sizes to be conclusive (S9 Table).

## RAS genes are prognostic biomarkers for AML

We then asked if the RAS gene expression could help prognosticate AML patients. For this purpose, we utilized the training set within the GSE12417 [30] dataset, including 163 samples of bone marrow or peripheral blood mononuclear cells from adult patients with untreated acute myeloid leukemia. Patients in this cohort were also-treated with TAD protocol which contains Daunorubicin, which is also used as the starting material for semi-synthetic manufacturing of Doxorubicin. We found that high expression of genes *IGF2R* and *CTSA* were both associated with better overall survival, while the opposite was true for *ATP6AP2* when patients were classified in either "High" or "Low" groups based upon LRMC cutoffs for each gene separately (see Materials and methods section) (Fig 2). We then stratified patients into "Good" and "Bad" prognosis groups using the best cutoff values obtained for these three genes as explained in the method section. As shown in Fig 3, it was revealed that there was a striking difference in overall survival in the groups that were predicted as "Good" and "Bad". The "Good" group showed better survival than the "Bad" group. Since the patients were all treated with Daunorubicin, these data suggest that the expression pattern of these genes was able to identify patients which are responders of this therapy.

## Discussion

RAS' local presence in the marrow affects the most important stages of physiological and pathological blood cell proliferation, and also has important roles in the development of blood cancers. It has been shown that RAS plays important roles in drug resistance to chemotherapeutic agent in addition to angiogenesis, invasion and proliferation [9, 24, 45–47]. Inevitably, most of these processes are interdependent. Most of the increased metastasis and invasion occurs due to an active RAS results in angiogenesis [45, 48, 49]. AT1R upregulation in ovarian cancer and increased expression of AT1R and ACE in prostate cancer, and AGTR1 in breast cancer; localized RAS presence in gastric cancer and its correlation with tumor spread and progression; demonstrate strong associations of RAS with various cancers. Irregularity of RAS components

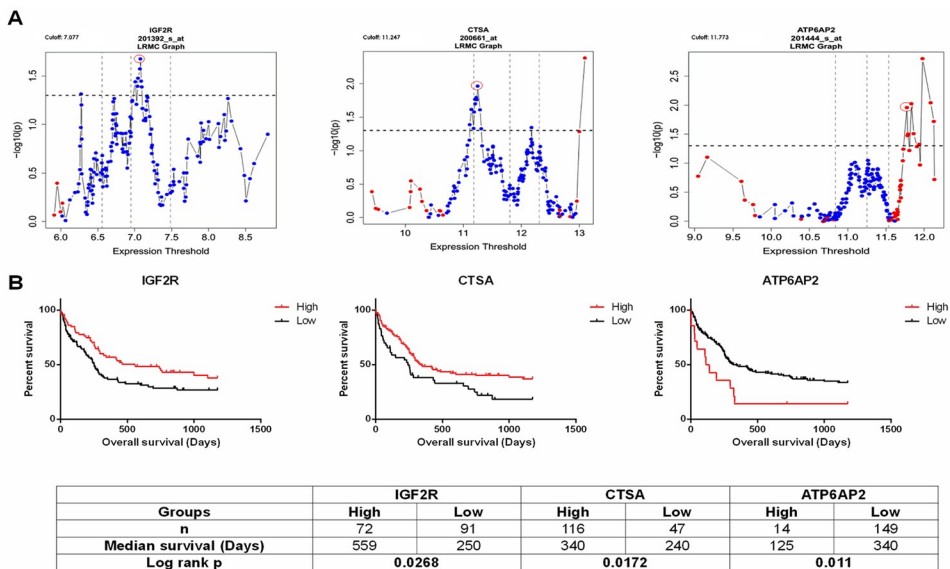

| | IGF2R | | CTSA | | ATP6AP2 | |
|---|---|---|---|---|---|---|
| **Groups** | High | Low | High | Low | High | Low |
| **n** | 72 | 91 | 116 | 47 | 14 | 149 |
| **Median survival (Days)** | 559 | 250 | 340 | 240 | 125 | 340 |
| **Log rank p** | 0.0268 | | 0.0172 | | 0.011 | |

**Fig 2. Log Rank Multiple Cutoff (LRMC) plots and Kaplan Meier curves for dataset GSE12417.** (A) LRMCs of IGF2R (Probeset: 201392_s_at), CTSA (Probeset: 200661_at) and ATP6AP2 (Probeset: 201444_s_at). Graphic shows log rank based p values in the y axis for the "high" and "low" expression groups generated by all possible expression based cutoffs shown on the x axis (for details see Materials and methods). HRs above one and below one are shown with red and blue colors for specific cutoffs, respectively. Vertical dotted lines show 25th, 50th and 75th percentiles and horizontal dotted line shows significance cutoff 0.05 (-log10(p) = 1.301). From LRMC graphs, we selected cutoffs 7.077, 11.247 and 11.773 for IGF2R, CTSA and ATP6AP2 respectively which are highlighted in figure with red circle. Patients were divided into high and low groups based on these cutoffs. (B) Kaplan Meier plots for patients classified in high and low expression based on LRMC cutoffs. Patients classifed in high expression group of IGF2R and CTSA showed better overall survival when compared with low expression group and high expression group of ATP6AP2 showed worse survival when compared with low expression group. For all survival plots, overall survival time is shown in days for 163 AML patients. Table at the bottom shows number of patients in each group, median survival for each group and Log rank p value for Kaplan Meier analysis.

in cancer is strongly associated with increased angiogenesis and metastasis, and these parameters are associated with poor prognosis [50–54].

Gene expression profiling has revealed various AML subtypes related to diagnosis, therapy response and prognosis [55, 56]. Although gene expression profiling has not yet been integrated into clinical practice, this is expected to happen in near future.

In our study, we focused on RAS genes and identified their association with Doxorubicin and Etoposide sensitivity. We also show that RAS genes can be used to stratify AML patients into groups with distinct prognoses. Similar to our findings, low expression of IGF2R in non-small cell liver cancer has been associated with poor prognosis and high expression in bladder cancer has been associated with good prognosis [57, 58]. Although high CTSA expression was associated with poorer outcome in breast ductal carcinoma *in situ*, it was also found to suppress invasion and metastasis of colorectal cancer, suggesting tissue-specific differential roles [59, 60]. Recent studies linked *ATP6AP2* up-regulation to the progression of glioma and colorectal cancer, due to its roles in aberrant activation of the Wnt/beta-catenin signalling pathway [61, 62]. *ATP6AP2* was also shown to be a key component of the pro-angiogenic/proliferative arm of the RAS, which plays a role in the growth and spread of endometrial cancer [63]. Compared to the presence in the lysosome, it is found more in the cell membrane. Thus, it is clear that in this way it induces TGF-beta pathway activation. *IGF2R* is located in the membrane of organelles and is responsible for the transport to lysosome, and its intracellular functions have

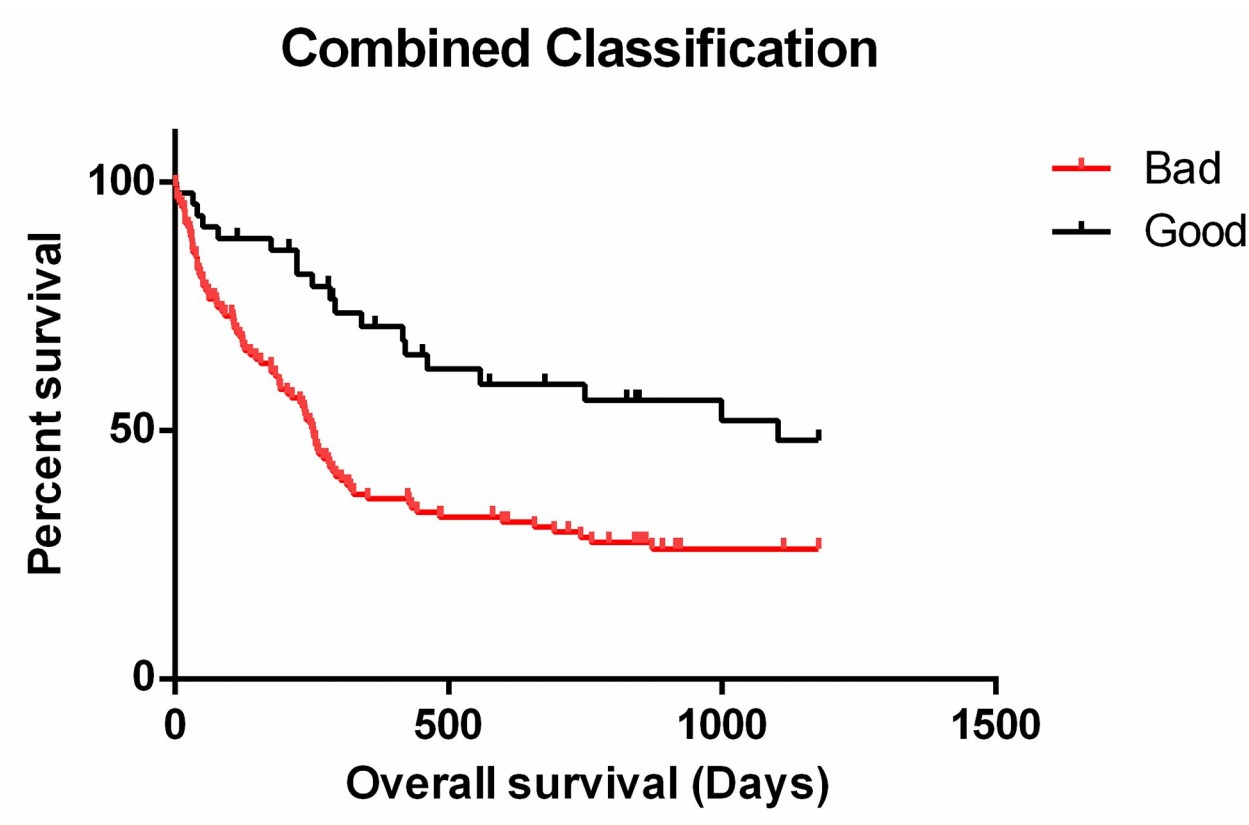

**Fig 3. Combined classification using IGF2R, CTSA and ATP6AP2 expression.** Patients were grouped as "Good" if they have high expression levels of IGF2R and CTSA and low expression levels of ATP6AP2 defined by expression value cutoffs in Fig 2. Rest of the patients were grouped as "Bad". Kaplan Meier plot shows "Good" group showed better survival when compared with "Bad" group as expected. Table at the bottom shows number of patients in each group, median survival for each group and Log rank p value for Kaplan Meier analysis.

not yet been clearly identified. *CTSA* is a protease found in the lysosome. The fact that these three genes function together in the lysosome suggests that lysosomal functions can contribute to cell sensitivity. *ATP6AP2* gene was found to cause disruption of V-ATPase formation and defects in the lysosomal glycosylation and autophagy [64]. Supportively, Doxorubicin has been reported to cause autophagy induced cell death in AML cells [65, 66].

GSEA revealed that sensitive cells were correlated with TNF-receptor interacting and response to type I IFN gene sets and resistant cells were correlated with regulation of TGF-beta production and FN-binding gene sets in AML, suggesting a mesenchymal phenotype.

A good and reliable subgrouping which can predict Doxorubicin sensitivity in AML was performed with the *ATP6AP2*, *IGF2R*, and *CTSA* gene combination. For those analyses, we utilized a Daunorubicin treated cohort, which is used as the starting material for semi-synthetic manufacturing of Doxorubicin. Therefore, the combination of these genes which can predict the sensitivity of Doxorubicin in AML patients may, therefore, be confirmed *ex vivo*.

The mutational analyses performed in this study had a small sample size with only two resistant cells. Therefore more conclusive results would be reached when this type of analysis is performed with larger sample sizes, or when mutational profiling is performed in patients treated with Doxorubicin, which may shed light on future studies.

## Conclusions

As a result, we identified *IGF2R*, *CTSA* and *ATP6AP2* gene biomarkers, which can subgroup AML patients into distinct good and bad prognostic groups. *ATP6AP2* was associated to resistance and *IGF2R* and *CTSA* were associated to sensitivity for Doxorubicin *in silico* and *in vitro*. In future studies, it is important to investigate whether these genes can be used for personalized treatment and improve the effectiveness of treatments.

## Supporting information

**S1 Fig. Hierarchical clustering of AML cell lines by sensitivity profiles for Doxorubicin and Etoposide.** The analysis reveals sensitive (six cell lines-green), intermediate (eight cell lines-orange) and resistant (three cell lines-red) subgroups for the 17 AML cell lines. Sensitivity to Doxorubicin and Etoposide is highly concordant in three subgroups. Green indicate low expression, orange indicate intermediate expression and red indicates high expression.
(TIF)

**S2 Fig. Comparative analysis of differentially enriched gene sets among drug sensitive and resistant cell lines.** (A) Plots showing gene sets enriched in sensitive cells, including genes interacting with TNF-receptor and genes affected in response to type I IFN stimulus. (B) Plots showing gene sets enriched in resistant cell lines, including genes having role in regulation of TGF-B production and genes interacting selectively and non-covalently with Fibronectin.
(TIF)

**S3 Fig. Expression levels of E-cadherin and Vimentin genes in AML cell lines.** RMA normalized gene expression values of CGP microarray data (y-axis) were used to determine EMT status of sensitive and resistant AML cell lines (x-axis) defined in S1 Fig. VIM: Vimentin (black bars), CDH1: E-cadherin (white bars). n.s. (not significant).
(TIF)

**S1 Table. Expression variance of RAS genes in AML cell lines.** RMA normalized gene expression values of 25 RAS genes were used to analyze variance, standard deviation (SD), mean and min-max difference among cell lines. Gene names is shown along with probe set ID.
(XLSX)

**S2 Table. Pearson correlation analysis between CGP IC50s and 6M IC50s.** IC50 values recalculated according to 6M approach using CGP raw cytotoxicity measurements were used

to calculate Pearson correlation analysis with CGP IC50 values. Strong correlations are observed for all drugs except for ATRA.
(PDF)

**S3 Table. Pearson correlation analysis between CGP gene expression data and CGP / 6M IC50s for 17 AML cell lines.** Eight genes expression data (nine probesets) was used to calculate correlation (as $R^2$ coefficient of determination) with IC50 values of four drugs (ATRA, Cytarabine, Etoposide, Doxorubicin) from CGP database along with recalculated data with 6M approach. Highlighted values with green and red indicate significant correlation in negative and positive manner respectively.
(PDF)

**S4 Table. Linear regression analysis between expression values of nine probesets and CGP / 6M IC50s for the discovery group (12 cell lines) and also for the ten times randomly divided different discovery groups (12 cell lines).** Adjusted $R^2$ values were calculated in Minitab 17 with eight genes (nine probesets) for four drugs using CGP gene expression data and CGP / 6M IC50 values of the 12 AML cell lines. High correlations are observed with Etoposide and Doxorubicin (bold, for Etoposide $R^2 > 90\%$, for Doxorubicin $R^2 > 80\%$). Averages of adjusted $R^2$ values of ten randomly divided groups were calculated in Minitab 17 with eight genes (nine probesets) for four drugs using CGP gene expression data and CGP / 6M IC50 values of the 12 AML cell lines. High correlation is observed with Doxorubicin and fine with Etoposide (bold, for Doxorubicin $R^2 > 85\%$, for Etoposide $R^2 > 60\%$). Asterisk represents the analysis in which Minitab could not perform linear regression analysis.
(PDF)

**S5 Table. Generation of linear regression models using CGP gene expression data and CGP / 6M IC50 data of the discovery group (12 AML cell lines) for drug sensitivity predictions.** (A) Individual genes and gene combinations were used to generate linear regression models using IC50 values of Doxorubicin and Etoposide from CGP and 6M IC50. Highest correlation is observed in *IGF2R/ATP6AP2/CTSA* combination with Doxorubicin CGP and 6M IC50 values. And, highest correlation is observed in *IGF2R/ATP6AP2/CTSA/CPA3* combination with Etoposide CGP and in *ANPEP/ATP6AP2/CTSA/CPA3/AGT* combination with Etoposide 6M IC50 values. (B) Regression formulas for gene panels with highest correlations.
(PDF)

**S6 Table. Relative expression values of *ATP6AP2, IGF2R (two probesets), CTSA, CPA3, AGT* and *ANPEP* genes in nine AML cell lines and Pearson's correlation analysis for six genes between CGP gene expression data and *in vitro* qRT-PCR gene expression data of nine AML cell lines.** (A) Expressions of all genes was normalized to GAPDH expression. (B) *ATP6AP2, IGF2R (two probesets), CPA3, AGT, and ANPEP* gene expression data obtained from CGP in silico and *in vitro* qRT-PCR expression data from nine cell lines show significant correlations with *in vitro* qRT-PCR expression data with the exception of CTSA.
(PDF)

**S7 Table. *In vitro* and predicted IC50s (from CGP and 6M IC50 linear regression formulas) of Doxorubicin and Etoposide for nine AML cell lines.** *In vitro* IC50 values were obtained from cell viability measurements of the cell lines that are treated with six different concentrations of Doxorubicin and Etoposide separately (20, 10, 2, 1, 0.2, 0.1 μM). Predicted IC50s were calculated using the four formulas generated (with CGP / 6M IC50s) in the linear regression analysis with the normalized gene expression data obtained from qRT-PCR.
(PDF)

**S8 Table. Gene sets enriched in (A) sensitive cell lines, and (B) resistant cell lines.**
(PDF)

**S9 Table. Mutational data of sensitive, intermediate and resistant groups of AML cell lines.** Seven genes which are mutated in at least three AML cell lines were included to examine the relationship between mutational status and drug sensitivity. Red: mutations that cause change in aminoacid sequence, grey: unkown status of aminoacid change, change at the DNA level; blue: wild type.
(PDF)

**S1 File.**
(XLSX)

## Author Contributions

**Data curation:** Seyhan Turk, Muhammad Waqas Akbar, Murat Isbilen, Secil Demirkol Canli.

**Formal analysis:** Seyhan Turk, Can Turk, Ali Osmay Gure.

**Investigation:** Seyhan Turk, Can Turk.

**Methodology:** Seyhan Turk, Can Turk, Muhammad Waqas Akbar, Murat Isbilen, Secil Demirkol Canli, Umit Yavuz Malkan, Mufide Okay, Gulberk Ucar, Nilgun Sayinalp, Ibrahim Celalettin Haznedaroglu, Ali Osmay Gure.

**Project administration:** Seyhan Turk, Ali Osmay Gure.

**Resources:** Seyhan Turk, Umit Yavuz Malkan, Mufide Okay.

**Software:** Seyhan Turk, Can Turk, Muhammad Waqas Akbar, Murat Isbilen, Secil Demirkol Canli, Ali Osmay Gure.

**Supervision:** Seyhan Turk, Gulberk Ucar, Nilgun Sayinalp, Ibrahim Celalettin Haznedaroglu.

**Validation:** Seyhan Turk, Gulberk Ucar, Nilgun Sayinalp, Ibrahim Celalettin Haznedaroglu, Ali Osmay Gure.

**Visualization:** Seyhan Turk, Gulberk Ucar, Ali Osmay Gure.

**Writing – original draft:** Seyhan Turk, Baris Kucukkaraduman, Ali Osmay Gure.

**Writing – review & editing:** Seyhan Turk, Can Turk, Baris Kucukkaraduman, Ali Osmay Gure.

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
