## [Decision Letter · Decision Letter 0]

7 May 2020

PONE-D-20-09590

Renin Angiotensin System Genes are Biomarkers for Personalized Treatment of Acute Myeloid Leukemia with Doxorubicin or Etoposide

PLOS ONE

Dear Dr. Turk,

Thank you for submitting your manuscript to PLOS ONE. After careful consideration, we feel that it has merit but does not fully meet PLOS ONE’s publication criteria as it currently stands. Therefore, we invite you to submit a revised version of the manuscript that addresses the points raised during the review process by both Reviewers.

We would appreciate receiving your revised manuscript by Jun 21 2020 11:59PM. To enhance the reproducibility of your results, we recommend that if applicable you deposit your laboratory protocols in protocols.io, where a protocol can be assigned its own identifier (DOI) such that it can be cited independently in the future. For instructions see: http://journals.plos.org/plosone/s/submission-guidelines#loc-laboratory-protocols

We look forward to receiving your revised manuscript.

Kind regards,

Francesco Bertolini, MD, PhD

Academic Editor

PLOS ONE

Journal Requirements:

2. Please provide additional information about each of the cell lines used in this work, including any quality control testing procedures (authentication, characterisation, and mycoplasma testing). For more information, please see " ext-link-type="uri" xlink:type="simple">http://journals.plos.org/plosone/s/submission-guidelines#loc-cell-lines."

3. Please provide the source, product number and any lot numbers of the doxorubicin and etoposide obtained for your study.”

4. Please note that PLOS does not permit references to “data not shown.” Authors should provide the relevant data within the manuscript, the Supporting Information files, or in a public repository. If the data are not a core part of the research study being presented, we ask that authors remove any references to these data.

Reviewers' comments:

Reviewer's Responses to Questions

**Comments to the Author**

1. Is the manuscript technically sound, and do the data support the conclusions?

Reviewer #1: Yes

Reviewer #2: Yes

2. Has the statistical analysis been performed appropriately and rigorously? 

Reviewer #1: No

Reviewer #2: Yes

3. Have the authors made all data underlying the findings in their manuscript fully available?

Reviewer #1: Yes

Reviewer #2: Yes

4. Is the manuscript presented in an intelligible fashion and written in standard English?

Reviewer #1: Yes

Reviewer #2: Yes

5. Review Comments to the Author

Reviewer #1: Turk and colleagues in their research article entitled “Renin Angiotensin System Genes are Biomarkers for Personalized Treatment of Acute Myeloid Leukemia with Doxorubicin or Etoposide” perform a series of analyses with the aim to to verify if RAS genes can be good predictors of the sensitivity of two chemoterapeutics. Their bioinformatic approach identifies a series of genes that have been, in this research, tested with in vitro experiment. Additionally, applying again a computational approach, the authors stratify a cohort of patients previously sequenced on the basis of the previously mentioned genes.

Although this research article is a good piece of work, I think that, in its current state, it is not suitable for publication but it can be potentially interesting if some modifications will be done to the analyses and to the manuscript.

The main concern here is about the methodology implied in the first computational section.

I please recommend to specifically indicate, particularly in the method section, if the workflow-analyses performed have been either applied in previous researches or are reported here for the first time. One example is in the “IC50 Calculation Methods” section: the six different models seem introduced by the authors for the first time while, in the result section (line 207) it is referred to them as the “NCBI proposed 6-model approach”, is it the same? Can the author add a reference to this?

Finally I suggest to be more consistent and clear with the numbers/genes along the text.

The major points are listed here.

In the “Data normalization and variance analysis” is there a reason why “the genes whose variance was above 0.8 SD of the mean” were chosen? Additionally this number is not the same of the results in which is reported “which showed high variation in expression and therefore, selected 9 probesets (8 genes) with standard deviation values above 0.9” (line 205), I would suggest to add a reference or better explain this method. I was wondering why the author did not consider to calculate and consider adjusted p-value for the genes selected.

In “linear regression analyses” section the authors need to better clarify the steps they followed during this methodology, I suggest either to insert some references or clarify the steps. Please also clarify if in this case all the genes or only the 8 were used.

Moreover, I wonder if the Pearson’s correlation was always applied on normal distribution of data, if this is not the case I would suggest a Spearman correlation test.

In the results section the authors refer to 6M data which have not been explained before in the method section, these data likely are deriving from the raw CGP after applying the six model approach, I would suggest to the authors to add this information in the method section.

I suggest to replicate the random division of the groups and test if the results are consistent with the one obtained here. Moreover in the method section there is no mention of such a random division, please, add it. If there is a reason why the division was not replicated, please, mention it.

In vitro experiment the genes used are six, and the primers reported in the table 1 are for seven genes. Please clarify this and explain the reason why the authors did not consider all the eight genes from the in silico workflow. Along the text it is not clear if the sub-groups of genes belong to the initial eight. Please refromulate the text in order to give a better explanation of these numbers and other numbers of genes.

My suggestion is to either reorganize the figures or change the captions: in Fig.3 there is no explanation of the three panels (A, B and C) and neither of the colors. Moreover, beside the Kaplan-Meier curves there are other 3 plots which are not explained. The same for Fig. 4.

Minor points:

- Line 79, when E-MTAB-783 is indicated, please cite the two research articles that contributed to produce these data:

1) Garnett MJ, Edelman EJ, Heidorn SJ, Greenman CD, Dastur A, Lau KW, Greninger P, Thompson IR, Luo X, Soares J, Liu Q. Systematic identification of genomic markers of drug sensitivity in cancer cells. Nature. 2012 Mar;483(7391):570-5.

2) Venkova L, Aliper A, Suntsova M, Kholodenko R, Shepelin D, Borisov N, Malakhova G, Vasilov R, Roumiantsev S, Zhavoronkov A, Buzdin A. Combinatorial high-throughput experimental and bioinformatic approach identifies molecular pathways linked with the sensitivity to anticancer target drugs. Oncotarget. 2015 Sep 29;6(29):27227.

The 1) is already present in the manuscript as ref number 32.

-Line 89, please insert the article “the” when referring to the 17 AML and to the 25 genes that are taken by CGP and are indicated in the results section. Moreover, consider to add this info also on the methods.

-Lines 121, please insert the website of Minitab 17

-Line123-124 if the authors are referring to the same eight genes that have been mentioned in the MM Normalization section I would suggest to point it out.

-Lines138 Please cite the reference or website for Cluster3.0 and Java Treeview software.

- The link at line 144 does not work, please indicate the number of pathways and the number of genes that were present in the C5_all Gene ontology database, and which version of the database was used.

-Line 188 if the script is available provide it as supplementary information or in a github repository

-Line 217 the authors are referring to 4 drugs and 8 genes, are these numbers and data the same that have been identified in the previously mentioned analyses? Why did the authors perform linear regression analyses at this step? Please report this information in this section of the manuscript and if the genes/drugs are the ones already mentioned add the definite article “the”.

-Lines 245-247 please refer to which correlation analysis the authors are referring to. Moreover, PLOS does not accept references to “data not shown.”

- Line 248 please indicate why only four formulas were applied and change 4 in four.

-Lines291-292 when the authors refer to “We then stratified patients using the best cut-off values obtained for these 3 genes” please add, “as explained in the method section”.

-Line 292 please substitute 3 with three

-Figure 1 A) and B) are not indicated. Define the Sy.x parameter, is it the value for the residuals? Please add this information also in the methods.

-Line562 please reformulate “ve resistant”

-Uniform the numbers, below 10 the number should be indicated as word.

Reviewer #2: Seyhan Turk and co-worker in their work demonstrate that expression of Renin-Angiotensin System (RAS)-related genes predict drug responses (Doxorubicin and Etoposide) in AML patients. Moreover, authors show that identified RAS genes expression stratify AML patients into different subtypes with distinct prognosis. Overall, presented data support use of RAS gene expression analysis as novel tool for AML drug-sensitivity and disease prognostication. Unfortunatley, altough an elegant set of in-silico approaches have been employed, the lack of experimental analyses with appropriate functional in-vitro and in-vivo represents the main drawback of the entire work. In detail:

Major points

• 17 AML cell lines included in CGP database have been chosen for in-silico analysis. In parallel, 9 AML cell lines have been testd for in vitro studies. Are those the same cells included in short list for in-silico analysis? Furthermore, did you see any differences based on their specific genetic background (mutational analysis)?

• Importantly, GEP analysis have been performed on genes, among those of RAS system, with higher expression variability. Why did you reject those with less variation for your analysis?

• Could you please detail the NCBI proposed 6-model used approach?

• To make in vitro date more consostent, could be useful including gene expression analysis as well as IC50 values for all tested AML cell lines.

• Data showed in Table 2 are not clear. Could you please describe it better?

• The prognostic role of 3-gene expression signature need deeper analysis. Why did you analyze only 3 genes? What about other RAS genes? Did you perform a cumulative analysis of RAS-related genes?

• As per Authors own admission, the major study limitation is lack of mutational data analysis. Indeed, it’s worth to be investigated AML patiens subclasses including those carryng poor prognostic mutations such as FLT3. To this aim a detailed description of used AML cell lines could help (i.e. carryng FLT3-ITD or WT, NPM1 etc.)

Minor

• Please pospone figures legend at the end of manuscript right after refernces

• The gene set enrichment analysis revealed findinds that are not supported by experimental data. Overall these data are somehow confusing because are not conclusive at all. In my opinion it’s better including these data as supplementary results to make conclusion more focused

• In the Materials and methods the first sentence of paraghraph is quite misleading. Additionally, please include reference for CGP database.

6. PLOS authors have the option to publish the peer review history of their article (what does this mean?). If published, this will include your full peer review and any attached files.

Reviewer #1: No

Reviewer #2: Yes: Michele Cea

---

## [Author Response · Author response to Decision Letter 0]

21 Jun 2020

Dear Editor,

We would like to thank the Editorial Board and the Referees for all of the important contributions, which will improve the paper. 

We have carefully reviewed the comments and revised the manuscript accordingly. Below please find the answers to the Editor’s and Reviewer’s comments. 

Yours faithfully,

Journal Requirements:

Comment 1.

Response 1. 

PLOS ONE's style requirements fully checked and revisions were done. 

Comment 2.

2. Please provide additional information about each of the cell lines used in this work, including any quality control testing procedures (authentication, characterization, and mycoplasma testing). For more information, please see http://journals.plos.org/plosone/s/submission-guidelines#loc-cell-lines."

Response 2.

Additional information is added for the used cell lines (see In vitro section in Materials and Methods). 

Comment 3.

3. Please provide the source, product number and any lot numbers of the doxorubicin and etoposide obtained for your study.”

Response 3.

Drug information has been added to the manuscript. 

Comment 4.

4. Please note that PLOS does not permit references to “data not shown.” Authors should provide the relevant data within the manuscript, the Supporting Information files, or in a public repository. If the data are not a core part of the research study being presented, we ask that authors remove any references to these data.

Response 4.

The “data not shown” was removed and the data was added as “S8 Table”.

Comments to the Author

Reviewer #1:

Comment 1.

The main concern here is about the methodology implied in the first computational section.

I please recommend to specifically indicate, particularly in the method section, if the workflow-analyses performed have been either applied in previous researches or are reported here for the first time. One example is in the “IC50 Calculation Methods” section: the six different models seem introduced by the authors for the first time while, in the result section (line 207) it is referred to them as the “NCBI proposed 6-model approach”, is it the same? Can the author add a reference to this? Finally, I suggest to be more consistent and clearer with the numbers/genes along the text.

Response 1.

Here, we report the 6-model (6M) approach for the first time which depends on a non-linear logistic regression function explained in NIH/NCGC assay guidelines. We derived six different versions of this function and select the one with the lowest error rate among all for the calculation of cytotoxicity values. References were added and the “IC50 Calculation Methods” section has been re-written. No inconsistency could be found with the numbers/genes given throughout the entire paper.

The major points are listed here.

Comment 2. 

In the “Data normalization and variance analysis” is there a reason why “the genes whose variance was above 0.8 SD of the mean” were chosen? Additionally, this number is not the same of the results in which is reported “which showed high variation in expression and therefore, selected 9 probesets (8 genes) with standard deviation values above 0.9” (line 205), I would suggest to add a reference or better explain this method. I was wondering why the author did not consider to calculate and consider adjusted p-value for the genes selected. 

Response 2.

In order to choose the genes which will be used in Real-time PCR for validations, we aimed to choose the most variable genes which could give detectable fold differences in vitro. The variance value of 0.8 was chosen arbitrarily. For these analyses, variance and standard deviation are the same values; we decided to use “variance”. The “Data normalization and variance analysis” section has been expanded and detailed. Since, we are here trying to choose the most variant genes, we did not calculate and consider adjusted p-value for the genes selected. 

Comment 3.

In “linear regression analyses” section the authors need to better clarify the steps they followed during this methodology, I suggest either to insert some references or clarify the steps. Please also clarify if in this case all the genes or only the 8 were used. 

Moreover, I wonder if the Pearson’s correlation was always applied on normal distribution of data, if this is not the case I would suggest a Spearman correlation test.

Response 3.

The linear regression analyses section has been re-written. Linear regression analysis was performed with the highly variant eight RAS genes and minimal gene combinations which are now explained more clearly.

We performed Pearson r correlation analysis throughout the paper as we consistently obtained better p values with it, as compared to Spearman’s test.

Comment 4.

In the results section the authors refer to 6M data which have not been explained before in the method section, these data likely are deriving from the raw CGP after applying the six model approach, I would suggest to the authors to add this information in the method section.

Response 4.

With 6M approach we recalculated IC50 values from raw CGP data for 17 AML cell lines treated with four drugs (ATRA, Cytarabine, Etoposide and Doxorubicin) using an in-house R script. We refer to this data as 6M IC50s.

Additionally, we treated 9 AML cell lines with Doxorubicin and Etoposide in vitro and their IC50 values were calculated using 6M approach, as well. We refer to this data as in vitro IC50s.

These are explained in the methods section.

Comment 5.

I suggest to replicate the random division of the groups and test if the results are consistent with the one obtained here. Moreover, in the method section there is no mention of such a random division, please, add it. If there is a reason why the division was not replicated, please, mention it.

Response 5.

In response to this comment we divided 17 cells randomly 10 times and generated 10 different discovery and test groups consisting of 12 cell lines and 5 cell lines, respectively. Linear regression models were generated in the 10 discovery groups separately. The 10 random models of sensitivity to Doxorubicin still have an average R2 above 85% for both CGP and 6M IC50, but R2 decreased slightly for models of sensitivity to Etoposide. Average R2 values now given in “S5 Table”. We added the 10 times random division to the method section and also mentioned it in the results section. However, the reason we performed our analyses without random divisions was because we wanted both the discovery and test cohorts to contain cells that spanned as large a sensitivity interval as possible. We therefore, present in this revised version results from both analyses.

Comment 6.

In vitro experiment the genes used are six, and the primers reported in the table 1 are for seven genes. Please clarify this and explain the reason why the authors did not consider all the eight genes from the in silico workflow. Along the text it is not clear if the sub-groups of genes belong to the initial eight. Please reformulate the text in order to give a better explanation of these numbers and other numbers of genes.

Response 6.

GAPDH was used as endogenous reference control. That’s why the primers are seven in the Table 1. 

We used six genes because in our linear regression analyses, highest correlation is observed in IGF2R/ATP6AP2/CTSA combination with Doxorubicin (both CGP and 6-model) and highest correlation is observed in IGF2R/ATP6AP2/CTSA/CPA3 combination with Etoposide (CGP) and ANPEP/ATP6AP2/CTSA/CPA3/AGT combination with Etoposide (6-model). Prediction model contains totally six genes for Doxorubicin and Etoposide. The section has been re-written for the sake of clarity.

Comment 7.

My suggestion is to either reorganize the figures or change the captions: in Fig.3 there is no explanation of the three panels (A, B and C) and neither of the colors. Moreover, beside the Kaplan-Meier curves there are other 3 plots which are not explained. The same for Fig. 4.

Response 7.

Figures and their legends were re-organized. 

Minor points:

Comment 8.

- Line 79, when E-MTAB-783 is indicated, please cite the two research articles that contributed to produce these data:

1) Garnett MJ, Edelman EJ, Heidorn SJ, Greenman CD, Dastur A, Lau KW, Greninger P, Thompson IR, Luo X, Soares J, Liu Q. Systematic identification of genomic markers of drug sensitivity in cancer cells. Nature. 2012 Mar;483(7391):570-5.

2) Venkova L, Aliper A, Suntsova M, Kholodenko R, Shepelin D, Borisov N, Malakhova G, Vasilov R, Roumiantsev S, Zhavoronkov A, Buzdin A. Combinatorial high-throughput experimental and bioinformatic approach identifies molecular pathways linked with the sensitivity to anticancer target drugs. Oncotarget. 2015 Sep 29;6(29):27227.

The 1) is already present in the manuscript as ref number 32.

Response 8.

Citations were added and the text was reorganized.

Comment 9.

-Line 89, please insert the article “the” when referring to the 17 AML and to the 25 genes that are taken by CGP and are indicated in the results section. Moreover, consider to add this info also on the methods.

Response 9.

In the method and results sections article "the" were inserted. Source of AML cell lines, CGP, was indicated. 

Comment 10.

-Lines 121, please insert the website of Minitab 17

Response 10.

The website was added.

Comment 11.

-Line123-124 if the authors are referring to the same eight genes that have been mentioned in the MM Normalization section, I would suggest to point it out.

Response 11.

These are the same genes, which is now indicated.

Comment 12.

-Lines138 Please cite the reference or website for Cluster3.0 and Java Treeview software.

Response 12.

The websites for software were added.

Comment 13.

- The link at line 144 does not work, please indicate the number of pathways and the number of genes that were present in the C5_all Gene ontology database, and which version of the database was used.

Response 13.

The reference was added. 

Dataset E-MTAB-783 has 22277 probesets IDs and these were collapsed into 13321 genes. For genes with more than one probeset, one with the highest expression was selected. “C5_all Gene ontology v6.1 database” was used for the analysis which has gene sets that contain genes annotated by the same GO term. Default filtering criteria in GSEA for gene sets is that it should have minimally 15 genes and maximally 500 genes. After applying this filter, analysis was performed for 4081 gene sets. This information was added to the method section, as well.

Comment 14.

-Line 188 if the script is available provide it as supplementary information or in a github repository.

Response 14.

The R script was provided as supplementary.

Comment 15.

-Line 217 the authors are referring to 4 drugs and 8 genes, are these numbers and data the same that have been identified in the previously mentioned analyses? Why did the authors perform linear regression analyses at this step? Please report this information in this section of the manuscript and if the genes/drugs are the ones already mentioned add the definite article “the”.

Response 15.

Yes, four drugs and eight genes are the same in the previously mentioned analyses. To examine the correlation of drug sensitivity with combined expression profile of the eight genes, linear regression analyses were performed. “Linear regression analyses” section was reviewed. Article “the” was added when needed.

Comment 16.

-Lines 245-247 please refer to which correlation analysis the authors are referring to. Moreover, PLOS does not accept references to “data not shown.”

Response 16.

This correlation analysis is referring to test the compatibility of in silico and in vitro gene expression data with each other. “data not shown” removed from the text and data now given in “S8 Table”. 

Comment 17.

- Line 248 please indicate why only four formulas were applied and change 4 in four.

Response 17.

We applied four formulas since there are four linear regression models for Doxorubicin (CGP and 6M IC50s) and Etoposide (CGP and 6M IC50s) with minimal gene lists, we used these four formulas to predict IC50 values for both drugs in test groups. 

“4” was changed to “four”. 

Comment 18.

-Lines291-292 when the authors refer to “We then stratified patients using the best cut-off values obtained for these 3 genes” please add, “as explained in the method section”.

Response 18.

The text was rewritten. 

Comment 19.

-Line 292 please substitute 3 with three

Response 19.

The number was indicated as “three”.

Comment 20.

-Figure 1 A) and B) are not indicated. Define the Sy.x parameter, is it the value for the residuals? Please add this information also in the methods.

Response 20.

A), B), C) and D) were indicated in the Figure 1. 

Sy.x is a standard deviation of the residuals. In our linear regression analyses the residual standard deviation used to describe the difference in standard deviations of CGP and 6M IC50s versus predicted IC50s. It is a goodness-of-fit measure used to show how well our predicted IC50s fit with the CGP and 6M IC50s. It is also mentioned in the methods section.

Comment 21.

-Line562 please reformulate “ve resistant”

Response 21. 

The text was reformulated.

Comment 22.

-Uniform the numbers, below 10 the number should be indicated as word.

Response 22.

Below 10 the number are indicated as word.

Reviewer #2:

Major points

Comment 1.

• 17 AML cell lines included in CGP database have been chosen for in-silico analysis. In parallel, 9 AML cell lines have been testd for in vitro studies. Are those the same cells included in short list for in-silico analysis? Furthermore, did you see any differences based on their specific genetic background (mutational analysis)?

Response 1.

Yes, they are same cell lines. 17 cell lines used in CGP data are listed in S1 Table. These are CTV-1, HL-60, GDM-1, HEL92.1.7, KASUMI-1, KMOE-2, K052, ML-2, MONO-MAC-6, NKM-1, NOMO-1, P31-FUJ, THP-1, QIMR-WIL, CMK, CESS, OCI-AML2. 

Cell lines studied in vitro experiments are shown in the method section “Cell lines and cytotoxicity experiments”. These are HEL92.1.7, KASUMI-3, GDM-1, QIMR-WIL, CESS, P31/FUJ, NOMO-1, KASUMI-1 and SKM-1. 

Seven cell lines used for in silico analyses were also used for in vitro experiments (HEL92.1.7, KASUMI-1, GDM-1, QIMR-WIL, CESS, P31/FUJ, NOMO-1), as stated in the manuscript. 

We performed a comprehensive mutational analysis in order to see if any mutational pattern overlaps with the sensitivity profile, and added the results in the manuscript. Also, nine AML cell lines were WT for NPM1 and FLT mutations. 

Comment 2.

• Importantly, GEP analysis have been performed on genes, among those of RAS system, with higher expression variability. Why did you reject those with less variation for your analysis?

Response 2.

In order to choose the genes that will be used further in Real-time PCR for validations, we first aimed to choose the most variable genes which are also highly likely to give detectable fold differences via PCR. 

Comment 3.

• Could you please detail the NCBI proposed 6-model used approach?

Response 3.

It is detailed in the “IC50 Calculation methods” section. 

NCBI methodology or "NIH/NCGC-proposed methodology" suggests a calculation methodology similar to 6-model approach. The function used to model the data is widely being used for cytotoxicity calculations. We derived different versions of this function, which is partly suggested by NIH/NCGC, and select the one with the lowest error rate among all for the calculation of cytotoxicity values. 

Comment 4.

• To make in vitro date more consistent, could be useful including gene expression analysis as well as IC50 values for all tested AML cell lines.

Response 4.

To make more consistent in vitro cytotoxicity IC50s, predicted IC50 and QPCR relative gene expression values are now given in “S7 Table” and “S9 Table”. 

Comment 5.

• Data showed in Table 2 are not clear. Could you please describe it better?

Response 5.

The data has been described in more detail.

Comment 6.

• The prognostic role of 3-gene expression signature need deeper analysis. Why did you analyze only 3 genes? What about other RAS genes? Did you perform a cumulative analysis of RAS-related genes?

Response 6.

First, we started analyzing the 25 RAS genes all together. After applying a variance cut-off, this number decreased to eight genes. After linear regression analysis, these eight genes decreased to six genes (four regression formulas contain totally six genes for Etoposide and Doxorubicin and for two different IC50s (CGP and 6M)). But three genes (IGF2R, CTSA, ATP6AP2) were common for all regression formulas except for one “Etoposide 6M”. Therefore, we analyzed that three genes’ biomarker potential. 

Even for Doxorubicin, only these three genes come out in the regression formulas with both CGP and 6M IC50s. Only one additional gene (CPA3) comes out in the regression formula for Etoposide CGP IC50s. Analyses of six genes was necessary to produce predicted IC50s from regression formulas.

Comment 7.

• As per Authors own admission, the major study limitation is lack of mutational data analysis. Indeed, it’s worth to be investigated AML patiens subclasses including those carryng poor prognostic mutations such as FLT3. To this aim a detailed description of used AML cell lines could help (i.e. carryng FLT3-ITD or WT, NPM1 etc.)

Response 7.

Since there is no mutational data in the patient dataset we could not perform mutational analyses with clinical data. However, we evaluated the mutational profile of 14 out of 17 AML cell lines used in in silico analyses. Seven genes (TP53, RBMX, NRAS, ANKRD36C, TNS1, TTN and ASXL1) which are mutated in at least three cell lines were included in mutational analysis and our results are given in “S4 Fig”. 

For FLT3 and NPM1 gene, none of the used in vitro AML cell lines were mutants.

Minor points

Comment 8.

• Please pospone figures legend at the end of manuscript right after refernces

Response 8.

PLOS Journal requirements state the following: Place figure captions in the manuscript text in read order, immediately following the paragraph where the figure is first cited. 

Comment 9.

• The gene set enrichment analysis revealed findinds that are not supported by experimental data. Overall these data are somehow confusing because are not conclusive at all. In my opinion it’s better including these data as supplementary results to make conclusion more focused.

Response 9.

The gene set enrichment analysis results have been changed as supplementary “S2 Fig” and “S10 Table”.

Comment 10.

• In the Materials and methods the first sentence of paraghraph is quite misleading. Additionally, please include reference for CGP database.

Response 10. 

Paragraph was re-written. The CGP database references were added.

---

## [Decision Letter · Decision Letter 1]

14 Jul 2020

PONE-D-20-09590R1

Renin angiotensin system genes are biomarkers for personalized treatment of acute myeloid leukemia with doxorubicin as well as etoposide

PLOS ONE

Dear Dr. Turk,

Thank you for submitting your manuscript to PLOS ONE. After careful consideration, we feel that it has merit but does not fully meet PLOS ONE’s publication criteria as it currently stands. Therefore, we invite you to submit a revised version of the manuscript that addresses the points raised during the review process, particularly by Reviewer #1.

We look forward to receiving your revised manuscript.

Kind regards,

Francesco Bertolini, MD, PhD

Academic Editor

PLOS ONE

Reviewers' comments:

Reviewer's Responses to Questions

**Comments to the Author**

1. If the authors have adequately addressed your comments raised in a previous round of review and you feel that this manuscript is now acceptable for publication, you may indicate that here to bypass the “Comments to the Author” section, enter your conflict of interest statement in the “Confidential to Editor” section, and submit your "Accept" recommendation.

Reviewer #1: (No Response)

Reviewer #2: All comments have been addressed

2. Is the manuscript technically sound, and do the data support the conclusions?

Reviewer #1: Partly

Reviewer #2: Yes

3. Has the statistical analysis been performed appropriately and rigorously? 

Reviewer #1: No

Reviewer #2: Yes

4. Have the authors made all data underlying the findings in their manuscript fully available?

Reviewer #1: Yes

Reviewer #2: Yes

5. Is the manuscript presented in an intelligible fashion and written in standard English?

Reviewer #1: No

Reviewer #2: Yes

6. Review Comments to the Author

Reviewer #1: I appreciate the revision of the methods and the extension of the non-linear regression model section, now the methods are sufficiently explained and more clear. However I have noticed that the new version of the manuscript has some parts that need to be reformulated, many typos and inconsistencies along the text.

Additionally it is missing the explanation of the statistics applied in some parts of the manuscript and my concern here is whether or not these tests were rigorously applied.

Finally, I suggest a revision of the Supplementary Materials provided and that the R scripts file will be provided.

For these reasons I am still not considering the manuscript suitable for publication in its current form but I suggest the following points to be addressed to be taken in consideration for publication.

Line 82 the authors should insert the references in the correct location, if the reference number 30 is referring to the GSE12417 dataset it should be inserted right after it. As this, there are other similar cases along the text, therefore, I please invite the authors to check this in the manuscript.

Line 91, please remove “were chosen arbitrarily”, if any other study used this parameter please cite it.

Line 99-100 “explained in De Lean et. al, which is also explained” there is a repetition of the word explain, please substitute with “reported”.

Line129 please, insert here that the IC50 values were included in the CGP. If I am not wrong the authors mentioned this already at line 243.

Line 130-131 Please provide the R script and refer to it as supplementary material.

Line 131 “We refer to this data as 6M IC50”, please be consistent with this nomenclature along the text, sometimes it is called “6M IC50” others “IC50 6M” other only “6M”

Line 135 “9 AML cell lines were treated with” the number 9 needs to be written in words. Additionally I suggest to reformulate or clarify the meaning of the sentence “IC50 values were calculated using the 6M approach in vitro.”, did the authors mean that the values were calculated using the 6M approach on the data obtained from the in vitro analysis?

Line 138 8 needs to be written as a word.

The correlations to which the authors are referring here is the one shown in S3 Table, how was this correlation calculated? If it was with graphpad I would suggest inserting a sentence at the end of this section saying that all the correlations were calculated with Graphpad software. Moreover, Pearson correlation should be used when both the variables are normally distributed; in the response to the reviewers the authors mentioned that they got better results with this method but I was wondering if the two variables were tested for normal distribution or not.

Line 151 please delete “for 10 models” at the end of the sentence, it is redundant.

Line 161 please substitute “that is used to describe” with “that here has been used to describe”.

Line 173-174 please correct “Gene set enrichment analyses was” with “Gene set enrichment analysis was”

Line 223 please correct “Clinical data was” with “Clinical data were”

Line 226 please, when the R script is mentioned in the text, refer to the supplementary material in which it is contained.

Lines 228-229 These lines need a reformulation. First, you should put as references the two studies (PMID: 31949498, PMID: 28607584), second, please change “ 'Low' = low expression” with “‘Low’ (low expression)” and “‘High = high expression’” with “‘High’ (high expression)” and finally, substitute “our previous studies” with “as in refX and refY”.

Line 242 I ask the authors to rewrite the citations when CGP database is mentioned.

Line248 the authors mention: “We observed strong correlations between IC50 values obtained from CGP”, as said above, this correlations need to be clarified in the MM section.

Line254 The sentence “We then asked whether combined expression analyses of

genes could correlate better with drug sensitivity data.” should be linked with the next paragraph.

Line257-262 please re-arrange these sentences because they are not clear. The explanation of the workflow used has some typos and english grammar errors. Moreover, some parts are already mentioned in the MM section and should be removed.

Line 264 please, remove “RAS genes” it is a redundancy; if it is not, I please ask the authors to explain why it is mentioned here.

TableS4 and S5 should be merged into the same file.

The name of the columns should be revised, precisely: on the top of the column referring to CGP please indicate CGP and on the top of the column referring to 6M IC50 indicate 6M IC50. I also suggest to name the sheets of the .xlsx table according to the table.

The two above mentioned indications are applicable also to the other S Tables.

Line277 6M IC50 needs to be indicated accordingly along the text.

TableS6, the columns referring to each group need to be marked.

Figure 1: The names in the title need to be consistent with the content of the text, therefore 6M needs to be substituted with 6M IC50.

Line278 I would suggest to report also here the name of the six genes that have been selected.

Line294 a comma between “ANPEP ATP6AP2” is missing.

I suggest the authors either merge table S7 and S8 or put them in different sheets of the same file.

S1 Fig: I suggest that the legend of this figure will also include the meaning of the colors. Maybe a scale of colors should be provided in the figure.

Line320-322 EMT acronym is not explained along the text and there is no mention of the statistical test involved when the authors say: “there were no significant differences between the two groups”. The authors need to mention the test performed (Wilcoxon or t-test according to the distribution of the data) and/or report it in the mm section.

S4 Fig needs to be included as a table or a different figure should be provided.

Line 334-335 I was wondering how the authors investigated the up-regulation of the three genes. Additionally, as previously mentioned from the reviewer 2, it is still not clear to me the choice of these three genes; if it is related to the fact that these genes were the one common for all regression formulas I suggest to mention it in the text and/or mm section.

Line332 I recommend to add again the reference for GSE12417 when it is mentioned.

Line334 Doxorubicin needs to be indicated with the first letter in uppercase.

Line 337 please add in the parenthesis together with Fig2 “see Materials and Methods section”.

Line 339-341 these lines need to be reformulated.

Line352 the word “cut-off” needs to be consistent along the text and the numbers need to be rounded.

Since many S tables are really small I suggest to insert them in a unique pdf file and leave as excel only those that do not fit on a pdf page.

The script file is missing, it should be provided in a .zip file with all the codes.

The section Acknowledgments is blank.

Reviewer #2: (No Response)

7. PLOS authors have the option to publish the peer review history of their article (what does this mean?). If published, this will include your full peer review and any attached files.

Reviewer #1: No

Reviewer #2: **Yes: **Michele Cea

---

## [Author Response · Author response to Decision Letter 1]

27 Aug 2020

Dear Editor,

We would like to thank the Editorial Board and the Reviewer for all the contributions. You can find all the responses and the necessary revisions based on the reviewer’s comments below.

Yours faithfully,

Dr. Seyhan TURK, PhD

Review Comments to the Author

Reviewer #1: I appreciate the revision of the methods and the extension of the non-linear regression model section, now the methods are sufficiently explained and more clear. However I have noticed that the new version of the manuscript has some parts that need to be reformulated, many typos and inconsistencies along the text.

Additionally it is missing the explanation of the statistics applied in some parts of the manuscript and my concern here is whether or not these tests were rigorously applied.

Finally, I suggest a revision of the Supplementary Materials provided and that the R scripts file will be provided.

For these reasons I am still not considering the manuscript suitable for publication in its current form but I suggest the following points to be addressed to be taken in consideration for publication.

Response:

Accordingly, all sections were reformulated. All typos and inconsistencies have been corrected along with the text. All necessary explanations of all applied statistics were added to the manuscript. 

All tests were applied rigorously. All Supplementary Materials were reviewed and R script file was provided as a new Supplementary (S2 Table).

Comment 1:

Line 82 the authors should insert the references in the correct location, if the reference number 30 is referring to the GSE12417 dataset it should be inserted right after it. As this, there are other similar cases along the text, therefore, I please invite the authors to check this in the manuscript.

Response 1:

The locations of all references have been checked and necessary insertions were done. 

Comment 2:

Line 91, please remove “were chosen arbitrarily”, if any other study used this parameter please cite it.

Response 2:

The “were chosen arbitrarily” were removed from the text. 

Comment 3:

Line 99-100 “explained in De Lean et. al, which is also explained” there is a repetition of the word explain, please substitute with “reported”.

Response 3:

The “explained” was substituted with “reported”.

Comment 4:

Line129 please, insert here that the IC50 values were included in the CGP. If I am not wrong the authors mentioned this already at line 243.

Response 4:

Text was reviewed. 

The “IC50 values that were also included in the raw CGP data” and 

“We used two versions of CGP data, one original CGP data, second is recalculated 6M IC50 data by R script” were added to the text. 

Comment 5:

Line 130-131 Please provide the R script and refer to it as supplementary material.

Response 5:

The R script was provided in the text and it was referred as a new “S2 Table”

Comment 6:

Line 131 “We refer to this data as 6M IC50”, please be consistent with this nomenclature along the text, sometimes it is called “6M IC50” others “IC50 6M” other only “6M”

Response 6:

“6M IC50” it was corrected in all necessary locations in the manuscript. 

Comment 7:

Line 135 “9 AML cell lines were treated with” the number 9 needs to be written in words. Additionally I suggest to reformulate or clarify the meaning of the sentence “IC50 values were calculated using the 6M approach in vitro.”, did the authors mean that the values were calculated using the 6M approach on the data obtained from the in vitro analysis?

Response 7:

The number 9 was written in words. 

The sentence reformulated as

“In addition, IC50 values were calculated using the 6M IC50 approach on the data obtained from in vitro analysis in which nine AML cell lines were treated with Doxorubicin and Etoposide” to clarify the meaning.

Comment 8:

Line 138 8 needs to be written as a word.

The correlations to which the authors are referring here is the one shown in S3 Table, how was this correlation calculated? If it was with graphpad I would suggest inserting a sentence at the end of this section saying that all the correlations were calculated with Graphpad software. Moreover, Pearson correlation should be used when both the variables are normally distributed; in the response to the reviewers the authors mentioned that they got better results with this method but I was wondering if the two variables were tested for normal distribution or not.

Response 8:

The number 8 was written in words. 

At the end of the section “all the correlations were calculated with Graphpad software” were added.

Pearson's correlation analysis was applied only on normally distributed data. We observed that there was no evidence to reject normality in any variables (p>0.05) except for CPA3 gene (marked in red). The Supplementary data was updated so that an Excel file for the normality test results was added as "Correlation Analyses Results - Kolmogorov Smirnov (KS).

Comment 9:

Line 151 please delete “for 10 models” at the end of the sentence, it is redundant.

Response 9:

The “for 10 models” was removed from the text.

Comment 10:

Line 161 please substitute “that is used to describe” with “that here has been used to describe”.

Response 10:

The “that is used to describe” was substituted with “that here has been used to describe”.

Comment 11:

Line 173-174 please correct “Gene set enrichment analyses was” with “Gene set enrichment analysis was”

Response 11:

The “Gene set enrichment analyses was” was corrected with “Gene set enrichment analysis was”

Comment 12:

Line 223 please correct “Clinical data was” with “Clinical data were”

Response 12:

The “Clinical data was” was corrected with “Clinical data were”

Comment 13:

Line 226 please, when the R script is mentioned in the text, refer to the supplementary material in which it is contained.

Response 13: 

It was referred to the “S2 Table” when the R script is mentioned in the text.

Comment 14:

Lines 228-229 These lines need a reformulation. First, you should put as references the two studies (PMID: 31949498, PMID: 28607584), second, please change “ 'Low' = low expression” with “‘Low’ (low expression)” and “‘High = high expression’” with “‘High’ (high expression)” and finally, substitute “our previous studies” with “as in refX and refY”.

Response 14:

The necessary references were added (for PMID: 31949498, PMID: 28607584). 

The “'Low' = low expression” was changed to “‘Low’ (low expression)” and “‘High = high expression’” was changed to “‘High’ (high expression)”

And the “our previous studies” was changed to “as in [38,39].” 

Comment 15:

Line 242 I ask the authors to rewrite the citations when CGP database is mentioned.

Response 15:

The citations were rewritten. 

Comment 16:

Line248 the authors mention: “We observed strong correlations between IC50 values obtained from CGP”, as said above, this correlations need to be clarified in the MM section.

Response 16:

The section were clarified in the MM section. 

The “We performed a Pearson r correlation analysis between CGP IC50s and 6M IC50s to test the compatibility and observed strong correlations between them for drugs widely used in AML (Cytarabine, Etoposide, Doxorubicin) but not for ATRA (S3 Table)” was added. 

Comment 17:

Line254 The sentence “We then asked whether combined expression analyses of

genes could correlate better with drug sensitivity data.” should be linked with the next paragraph.

Response 17:

The sentences were linked together. 

Comment 18: 

Line257-262 please re-arrange these sentences because they are not clear. The explanation of the workflow used has some typos and english grammar errors. Moreover, some parts are already mentioned in the MM section and should be removed.

Response 18:

The section was rearranged.

Comment 19:

Line 264 please, remove “RAS genes” it is a redundancy; if it is not, I please ask the authors to explain why it is mentioned here.

Response 19:

The “RAS genes” were removed. 

Comment 20:

TableS4 and S5 should be merged into the same file.

The name of the columns should be revised, precisely: on the top of the column referring to CGP please indicate CGP and on the top of the column referring to 6M IC50 indicate 6M IC50. I also suggest to name the sheets of the .xlsx table according to the table.

The two above mentioned indications are applicable also to the other S Tables.

Response 20:

The S4 Table and S5 Table were merged into the same file as a new “S5 Table”. 

The name of the columns was revised. CGP and 6M IC50 were indicated on the top of the columns.

The sheets of the .xlsx tables were named according to the tables. 

Other S tables were also reviewed.

Comment 21:

Line277 6M IC50 needs to be indicated accordingly along the text.

Response 21:

“6M IC50” was indicated accordingly along the manuscript.

Comment 22:

TableS6, the columns referring to each group need to be marked.

Response 22:

S6 Table was revised accordingly. 

Comment 23:

Figure 1: The names in the title need to be consistent with the content of the text, therefore 6M needs to be substituted with 6M IC50.

Response 23:

The names in the titles of the Fig1 were renewed. 

Comment 24: 

Line278 I would suggest to report also here the name of the six genes that have been selected.

Response 24: 

The gene names were also reported in the text.

Comment 25:

Line294 a comma between “ANPEP ATP6AP2” is missing.

Response 25: 

The comma was added. 

Comment 26:

I suggest the authors either merge table S7 and S8 or put them in different sheets of the same file.

Response 26: 

Accordingly, Table S7 and Table S8 were merged as a new “S7 Table”. 

Comment 27:

S1 Fig: I suggest that the legend of this figure will also include the meaning of the colors. Maybe a scale of colors should be provided in the figure.

Response 27:

A color scale was added to the S1 Fig. And the meanings of the colors were added to the legend. 

Comment 28:

Line320-322 EMT acronym is not explained along the text and there is no mention of the statistical test involved when the authors say: “there were no significant differences between the two groups”. The authors need to mention the test performed (Wilcoxon or t-test according to the distribution of the data) and/or report it in the mm section.

Response 28:

The EMT acronym was explained in the text. The citations were added. We used t-test for statistical test. Accordingly, t-test was mentioned in the text. 

Comment 29:

S4 Fig needs to be included as a table or a different figure should be provided.

Response 29:

S4 Fig was included as a new “S10 Table”. 

Comment 30:

Line 334-335 I was wondering how the authors investigated the up-regulation of the three genes. Additionally, as previously mentioned from the reviewer 2, it is still not clear to me the choice of these three genes; if it is related to the fact that these genes were the one common for all regression formulas I suggest to mention it in the text and/or mm section.

Response 30:

Accordingly, the sentence was corrected as:

“We found that high expression of genes IGF2R and CTSA were both associated with better overall survival, while the opposite was true for ATP6AP2 when patients were classified in either “High” or “Low” groups based upon LRMC cutoffs for each gene separately (see Materials and Methods section) (Fig 2).”

The High expression and the opposite were determined using Log-rank with multiple cut-offs (LRMCs) algorithm as described in methods sections under the heading of clinical data validation. LRMC generates all possible cutoffs with their respective p values associated with Hazard ratios (Fig 2A). So for each gene, these cutoffs were generated and one cutoff with the smallest p-value was selected. Patients with expression values above this cutoff were labeled high and the rest were labeled as low. And for these groups, Kaplan Meier graphs were generated (Fig 2B). This explanation is also given in Fig 2 legend along with cutoff values as well.

The reason why these three genes are selected, as explained previously, is because they are common for both Doxorubicin and Etoposide except for Etoposide 6M IC50. As suggested by the reviewer the required explanation has been added to the “Clinical data validation-Log rank with multiple cutoffs (LRMC) and survival analysis” in the MM section.

Comment 31:

Line332 I recommend to add again the reference for GSE12417 when it is mentioned.

Response 31:

The reference for GSE12417 was added when needed along with the manuscript. 

Comment 32:

Line334 Doxorubicin needs to be indicated with the first letter in uppercase.

Response 32: 

The correction was done.

Comment 33:

Line 337 please add in the parenthesis together with Fig2 “see Materials and Methods section”.

Response 33:

The “see Materials and Methods section” was added. 

Comment 34:

Line 339-341 these lines need to be reformulated.

Response 34:

The lines were reformulated.

Comment 35:

Line352 the word “cut-off” needs to be consistent along the text and the numbers need to be rounded.

Response 35: 

All “cutoff” terms were made consistent and the numbers were rounded. 

Comment 36:

Since many S tables are really small I suggest to insert them in a unique pdf file and leave as excel only those that do not fit on a pdf page.

Response 36:

Except for S1 Table, all Supplementary Tables were converted to PDF files. 

Comment 37:

The script file is missing, it should be provided in a .zip file with all the codes.

Response 37:

The R Script file has only “one code” and it was provided in the new “S2 Table”.

Comment 38:

The section Acknowledgments is blank.

Response 38:

Accordingly, the title was deleted because this section is empty.

---

## [Decision Letter · Decision Letter 2]

11 Sep 2020

PONE-D-20-09590R2

Renin angiotensin system genes are biomarkers for personalized treatment of acute myeloid leukemia with doxorubicin as well as etoposide

PLOS ONE

Dear Dr. Turk,

Thank you for submitting your manuscript to PLOS ONE. After careful consideration, we feel that it has merit but does not fully meet PLOS ONE’s publication criteria as it currently stands. Therefore, we invite you to submit a revised version of the manuscript that addresses the points raised during the review process by Reviewer #1.

We look forward to receiving your revised manuscript.

Kind regards,

Francesco Bertolini, MD, PhD

Academic Editor

PLOS ONE

Reviewers' comments:

Reviewer's Responses to Questions

**Comments to the Author**

1. If the authors have adequately addressed your comments raised in a previous round of review and you feel that this manuscript is now acceptable for publication, you may indicate that here to bypass the “Comments to the Author” section, enter your conflict of interest statement in the “Confidential to Editor” section, and submit your "Accept" recommendation.

Reviewer #1: (No Response)

Reviewer #2: All comments have been addressed

2. Is the manuscript technically sound, and do the data support the conclusions?

Reviewer #1: Partly

Reviewer #2: Yes

3. Has the statistical analysis been performed appropriately and rigorously? 

Reviewer #1: Yes

Reviewer #2: Yes

4. Have the authors made all data underlying the findings in their manuscript fully available?

Reviewer #1: Yes

Reviewer #2: Yes

5. Is the manuscript presented in an intelligible fashion and written in standard English?

Reviewer #1: No

Reviewer #2: Yes

6. Review Comments to the Author

Reviewer #1: Turks and colleagues addressed most of the points raised during the last round of revision but one of the most important points, together with some typos/inconsistencies have not been corrected and/or introduced. In order to be accepted I recommend precisely covering all the points raised and to check for possible mistakes newly introduced.

Main point: the R script/scripts are still not included in the current version of the manuscript therefore I kindly ask to provide them in one of the following manners: either as file .R / .Rmd in a compressed file (.zip, gzip, tar.gz, ecc ecc) or as a link to a public repository. Currently the only file provided is a 1 x 1 table with the name of the script “SixModelIC50 V3.r” which does not include any code line.

Minor points to be addressed:

Line 86-87 It is not clear which dataset has been used for data normalization and variance analysis, the name “CGP microarray” combines the “CGP gene expression data” and the “microarray dataset GSE12417”. I please suggest, if the authors intend the “CGP gene expression data”, to use this name.

Line 103 in the new version of the manuscript it comes out that both the models and data have the same name “6M IC50”. This intent was not clear from the previous version, since there was a little bit of confusion in the names. Therefore, I suggest to use two different names for model and data (maybe using lowercase letters in one of the two or only 6M when referring to the model while 6M IC50 when referring to the data). Plase, be consistent along the text with this nomenclature when referring to one or to the other.

line 131-133: “. We referred to this data as 6M IC50. We used two versions of CGP data, one original CGP data, second is recalculated 6M IC50 data by this script.” Please, reformulate this sentence, it does not seem in the right place and it is not clear. I ask the authors to be consistent with the nomenclature and terms used in the other section.

138-139 the sentence “and observed strong correlations between them for drugs widely used in AML (Cytarabine, Etoposide, Doxorubicin) but not for ATRA (S3 Table).” needs to be moved in the result section. Plus there is a conflict on what it is mentioned in lines 254-256 of the results: “ Using t-test we observed strong correlations between CGP IC50 and 6M IC50, for drugs widely used in AML (Cytarabine, Etoposide, Doxorubicin) but not for ATRA (S3 Table).” The authors need to clarify if they used a t-test or a correlation Pearson test. The table is referring to the Pearson correlation.

Line 144: this is the first time the authors refer to the “CGP 6M IC50 data”, please be consistent with the nomenclature as mentioned in the previous paragraph and revision. If the name was only 6M IC50 it needs to be like this, otherwise if a new name is introduced, it needs to be specified before and the authors should explain it.

Line 232 It is quoted another R script that is not the same as the one used to calculate the 6M model but it is referred to as the same. I please ask the authors to correct this, and if it is available, to also provide this script together with the previous one. They can be put together in a compressed (.zip, gzip, tar.gz, ecc ecc) file.

Line 227-228 these new inserted lines need to be reformulated. “IGF2R, CTSA, ATP6AP2 were selected for clinical correlation studies is because they are common for all regression formulas except for Etoposide 6M IC50.” Likely “is because” is a typo, and this is the first time that the authors indicate “Etoposide 6M IC50”, what are they referring to?

Line 302 I kindly ask the authors to revise the use of the article “the” in this location. Have these cells been previously indicated in the text of the results? I suggest to remove the “the” and add (see Materials and Methods section) if the authors agree.

Lines 324-326 the authors should clarify how they “ tested E-Cadherin (epithelial marker) and Vimentin (mesenchymal marker) expression in silico”, if this analysis is referring to the S3 Fig, I please the authors to add at the end of this paragraph (S3 Fig). Moreover, the following paragraph (Lines 332-334) should not be separated if the authors are agreed. Finally the t-test is not shown in the S3 Fig and therefore the quote “(S3 Fig)” should be removed from line 334.

Fig 2 I please ask the authors to explain also the meaning of the red circle in the figure legend. Moreover, I think that the panel A legend needs to be more clear: I find it difficult to read it and it is not explicative of the figure.

Reviewer #2: Authors have addressed all my concerns thus making manuscript suitable for publication in its present form

7. PLOS authors have the option to publish the peer review history of their article (what does this mean?). If published, this will include your full peer review and any attached files.

Reviewer #1: No

Reviewer #2: No

---

## [Author Response · Author response to Decision Letter 2]

12 Oct 2020

Review Comments to the Author

Reviewer #1: Turks and colleagues addressed most of the points raised during the last round of revision but one of the most important points, together with some typos/inconsistencies have not been corrected and/or introduced. In order to be accepted I recommend precisely covering all the points raised and to check for possible mistakes newly introduced.

Main point: the R script/scripts are still not included in the current version of the manuscript therefore I kindly ask to provide them in one of the following manners: either as file .R / .Rmd in a compressed file (.zip, gzip, tar.gz, ecc ecc) or as a link to a public repository. Currently the only file provided is a 1 x 1 table with the name of the script “SixModelIC50 V3.r” which does not include any code line.

Response:

R Scripts were included in the manuscript as a link to a public Github Repository. “S2 Table was removed” 

Minor points to be addressed:

Comment 1.

Line 86-87 It is not clear which dataset has been used for data normalization and variance analysis, the name “CGP microarray” combines the “CGP gene expression data” and the “microarray dataset GSE12417”. I please suggest, if the authors intend the “CGP gene expression data”, to use this name.

Response 1.

“CGP microarray” was changed to “CGP gene expression data”. 

Comment 2.

Line 103 in the new version of the manuscript it comes out that both the models and data have the same name “6M IC50”. This intent was not clear from the previous version, since there was a little bit of confusion in the names. Therefore, I suggest to use two different names for model and data (maybe using lowercase letters in one of the two or only 6M when referring to the model while 6M IC50 when referring to the data). Plase, be consistent along the text with this nomenclature when referring to one or to the other.

Response 2.

As suggested by the reviewer, we used “6M” when referring “model” and we used “6M IC50” when referring “data”.

Comment 3.

line 131-133: “. We referred to this data as 6M IC50. We used two versions of CGP data, one original CGP data, second is recalculated 6M IC50 data by this script.” Please, reformulate this sentence, it does not seem in the right place and it is not clear. I ask the authors to be consistent with the nomenclature and terms used in the other section.

Response 3.

We reformulated and relocated the line. 

Comment 4.

138-139 the sentence “and observed strong correlations between them for drugs widely used in AML (Cytarabine, Etoposide, Doxorubicin) but not for ATRA (S3 Table).” needs to be moved in the result section. Plus there is a conflict on what it is mentioned in lines 254-256 of the results: “ Using t-test we observed strong correlations between CGP IC50 and 6M IC50, for drugs widely used in AML (Cytarabine, Etoposide, Doxorubicin) but not for ATRA (S3 Table).” The authors need to clarify if they used a t-test or a correlation Pearson test. The table is referring to the Pearson correlation.

Response 4.

We removed the “and observed strong correlations between them for drugs widely used in AML (Cytarabine, Etoposide, Doxorubicin) but not for ATRA (S3 Table)” from the Materials and Methods section. 

“T-test” was changed to “Pearson correlation” in the Results section. 

Comment 5.

Line 144: this is the first time the authors refer to the “CGP 6M IC50 data”, please be consistent with the nomenclature as mentioned in the previous paragraph and revision. If the name was only 6M IC50 it needs to be like this, otherwise if a new name is introduced, it needs to be specified before and the authors should explain it.

Response 5. 

“CGP 6M IC50 data” was changed to “6M IC50 data”.

Comment 6.

Line 232 It is quoted another R script that is not the same as the one used to calculate the 6M model but it is referred to as the same. I please ask the authors to correct this, and if it is available, to also provide this script together with the previous one. They can be put together in a compressed (.zip, gzip, tar.gz, ecc ecc) file.

Comment 6.

R Scripts were included in the manuscript as a link to a public Github Repository. “S2 Table was removed” 

Comment 7.

Line 227-228 these new inserted lines need to be reformulated. “IGF2R, CTSA, ATP6AP2 were selected for clinical correlation studies is because they are common for all regression formulas except for Etoposide 6M IC50.” Likely “is because” is a typo, and this is the first time that the authors indicate “Etoposide 6M IC50”, what are they referring to?

Response 7.

We reformulated the line.

Comment 8.

Line 302 I kindly ask the authors to revise the use of the article “the” in this location. Have these cells been previously indicated in the text of the results? I suggest to remove the “the” and add (see Materials and Methods section) if the authors agree.

Response 8.

We removed the “the” and added “(see Materials and Methods section)”. 

Comment 9.

Lines 324-326 the authors should clarify how they “ tested E-Cadherin (epithelial marker) and Vimentin (mesenchymal marker) expression in silico”, if this analysis is referring to the S3 Fig, I please the authors to add at the end of this paragraph (S3 Fig). Moreover, the following paragraph (Lines 332-334) should not be separated if the authors are agreed. Finally the t-test is not shown in the S3 Fig and therefore the quote “(S3 Fig)” should be removed from line 334.

Response 9.

We compared E-Cadherin (epithelial marker) and Vimentin (mesenchymal marker) expression using t-test. We clarified it and showed t-test p value on the S3 Fig.

We added “(S3 Fig)” at the end of paragraph and combined with the next paragraph. 

Comment 10.

Fig 2 I please ask the authors to explain also the meaning of the red circle in the figure legend. Moreover, I think that the panel A legend needs to be more clear: I find it difficult to read it and it is not explicative of the figure.

Response 10.

We simplified the legend, since we already have explanations for this method in methods section and we cited two previous studies. We also explained the meaning of red circle in the legend.

---

## [Decision Letter · Decision Letter 3]

4 Nov 2020

Renin angiotensin system genes are biomarkers for personalized treatment of acute myeloid leukemia with doxorubicin as well as etoposide

PONE-D-20-09590R3

Dear Dr. Turk,

We’re pleased to inform you that your manuscript has been judged scientifically suitable for publication and will be formally accepted for publication once it meets all outstanding technical requirements.

Kind regards,

Francesco Bertolini, MD, PhD

Academic Editor

PLOS ONE

Additional Editor Comments (optional):

Reviewers' comments:

Reviewer's Responses to Questions

**Comments to the Author**

1. If the authors have adequately addressed your comments raised in a previous round of review and you feel that this manuscript is now acceptable for publication, you may indicate that here to bypass the “Comments to the Author” section, enter your conflict of interest statement in the “Confidential to Editor” section, and submit your "Accept" recommendation.

Reviewer #1: All comments have been addressed

Reviewer #2: All comments have been addressed

2. Is the manuscript technically sound, and do the data support the conclusions?

Reviewer #1: (No Response)

Reviewer #2: Yes

3. Has the statistical analysis been performed appropriately and rigorously? 

Reviewer #1: (No Response)

Reviewer #2: Yes

4. Have the authors made all data underlying the findings in their manuscript fully available?

Reviewer #1: (No Response)

Reviewer #2: Yes

5. Is the manuscript presented in an intelligible fashion and written in standard English?

Reviewer #1: (No Response)

Reviewer #2: Yes

6. Review Comments to the Author

Reviewer #1: Turk and colleagues substantially improved the work, and I found that the manuscript in its current state can be suitable for publication in PLOSONE journal.

I only recommend that just a few very minor typos should be corrected before the final acceptance or the publication of this manuscript:

Line 179: the number "2227" needs to be indicate as 2,227.

Lines 268-270: please reformulate the sentence.

Line 648: the sentence is starting with “and”, I please the authors to correct this.

I please the authors be consistent with the name “CGP I50” when referring to it also in the Supp Tables (i.e: S3, S7 Tables).

Reviewer #2: The additional editing performed by Authors have made manuscript fully suitable for publication in PlosOne journal

7. PLOS authors have the option to publish the peer review history of their article (what does this mean?). If published, this will include your full peer review and any attached files.

Reviewer #1: No

Reviewer #2: No

---

## [Editor Report · Acceptance letter]

6 Nov 2020

PONE-D-20-09590R3 

Renin angiotensin system genes are biomarkers for personalized treatment of acute myeloid leukemia with doxorubicin as well as etoposide 

Dear Dr. Turk:

I'm pleased to inform you that your manuscript has been deemed suitable for publication in PLOS ONE. Congratulations! Your manuscript is now with our production department. 

Kind regards, 

on behalf of

Dr. Francesco Bertolini 

Academic Editor

PLOS ONE